# Echoes within the Reasoning: Stealth and Effective Watermarking via Chain of Thought

**Jiacheng Lu** [1]  **Yiming Li** [2]  **Tao Song** [1]  **Weijian Wang** [1]  **Wenjie Qu** [3]  **Haibing Guan** [1][†]  **Jiaheng Zhang** [3]

## Abstract

Large Language Models with Chain-of-Thought reasoning capabilities represent valuable intellectual property, yet existing black-box watermarking methods often trade robustness for reasoning fidelity by perturbing final answers or relying on fragile trigger patterns. We propose BiCoT, a watermarking framework that embeds ownership signals into the internal geometry of reasoning traces by aligning high-saliency structural anchors with a private signature subspace while regularizing ordinary control tokens to preserve semantic capacity. This design couples the watermark with reasoning-relevant representations, making removal difficult without disrupting the features that support coherent reasoning. To enable verification under model theft and representation drift, we introduce Robust Subspace Registration(RSR), a Top-$k$ logprob-based black-box verifier that uses sentinel tokens to calibrate systematic shifts in the output distribution. Experiments show that BiCoT preserves reasoning fidelity across diverse complex reasoning tasks while achieving robust detection under fine-tuning, quantization, model-level perturbations, and adaptive output-level attacks across in-domain and out-of-distribution settings. Code is available at https://github.com/JackLo111/BiCoT.

## 1. Introduction

Large Language Models (LLMs) have emerged as powerful reasoning engines, driving innovation across industries (Wang et al., 2024c; 2026; Xia et al., 2025). Currently, the immense computational resources and high-quality data required for their training render them high-value intellectual property (Wang et al., 2024a; Shao et al., 2025a; Li et al., 2025c; Lu et al., 2024). Consequently, these models face severe threats of unauthorized theft and unlicensed commercialization (Zhao et al., 2025; Shao et al., 2026).

To secure open-source models, the research community has pivoted towards model-based watermarking, which embeds ownership signatures directly into model weights (Liang et al., 2026). We categorize existing approaches into *white-box* and *black-box* paradigms based on the verification mechanism (Yuan et al., 2025; Gloaguen et al., 2025). While white-box methods require parameter access, black-box methods verify ownership via API queries (Yamabe et al., 2025; Xiong et al., 2026; Yang et al., 2025). Our focus is on the latter, which primarily consists of backdoor-based and model editing-based approaches, the main two categories.

Backdoor-based watermarking injects specific trigger-action patterns into the model (Li et al., 2023; Shao et al., 2025b; Yang et al., 2026). While intended to be stealthy, forcing the model to overfit to a small set of fixed triggers introduces critical flaws (Hu et al., 2024; Zhu et al., 2025; Li et al., 2025d). This paradigm fundamentally compromises *fidelity* on complex tasks and degrades *generalization* due to the interference of the backdoor mechanism (Ge et al., 2025). Furthermore, the reliance on fixed patterns renders the watermark susceptible to detection and filtering, and verification becomes unreliable if triggers are leaked or spoofed (Wang et al., 2019b; Xu et al., 2024b). To address these limitations, model editing-based methods leverage knowledge updating algorithms to inject watermarks as specific facts (Li et al., 2025b; Yue et al., 2025). Although these methods improve efficiency by avoiding full fine-tuning, they treat the watermark as static knowledge to be memorized. This approach is inherently fragile, as it relies on rigid question templates that are vulnerable to distribution shifts or adaptive attacks on prompt structures (Cheng et al., 2023; Wang et al., 2024b).

Crucially, both paradigms share a fundamental limitation: they verify ownership by modifying the *Final Answer*. This creates an unavoidable conflict with model utility, resulting in a zero-sum game between correctness and watermarking strength. This raises a fundamental question: *Does there exist a paradigm that achieves both stealthiness and*

---

[1]School of Computer Science, Shanghai Jiao Tong University, Shanghai, China [2]Nanyang Technological University,Singapore [3]National University of Singapore,Singapore. Correspondence to: Haibing Guan <hbguan@sjtu.edu.cn>.

*Proceedings of the 43rd International Conference on Machine Learning*, Seoul, South Korea. PMLR 306, 2026. Copyright 2026 by the author(s).

*effectiveness, while strictly preserving the model's fidelity?*

Luckily, the answer is **YES**! Inspired by (Guo et al., 2025), we identify the Chain-of-Thought (CoT) as a superior verification space that is functionally decoupled from the final answer. Our revisit of the CoT generation process reveals a distinct causal hierarchy. First, unlike the final answer which collapses into a low-entropy state, the reasoning trace provides a high-capacity semantic buffer. Second, tokens within this trace are unequal: gradient saliency demonstrates that *digit tokens* act as rigid "structural anchors" in GSM8K-style mathematical reasoning, whereas linguistic connectors are flexible. More generally, BiCoT treats anchors as a saliency-defined mask rather than a fixed token class; in other domains, logical connectives, legal predicates, or programming syntax may serve as anchors. Crucially, we discover a low-to-moderate injection regime where perturbations to these anchors propagate approximately linearly to the output before saturating into a plateau. This implies that ownership signals can be injected into the reasoning logic without severe semantic catastrophic interference, resolving the conflict between watermarking strength and fidelity.

Motivated by the insight that reasoning reliability hinges on a sparse set of highly "structural anchors," we argue that an effective watermark must inhabit this high-saliency subspace rather than the noise-tolerant margins. Consequently, we propose **BiCoT**, a novel method that embeds ownership directly into the intrinsic geometry of the reasoning process via manifold alignment. BiCoT establishes a functional entanglement between the watermark and logical reasoning. Specifically, we formulate watermarking as a bi-level optimization problem: the objective constrains the hidden states of structural anchors to collapse onto a proprietary signature subspace, while enforcing orthogonality for pure control tokens to preserve semantic capacity. Immediate post-anchor tokens are treated as a separate causal-halo category: they may exhibit short-range residual alignment, but this effect decays rapidly and does not imply that ordinary text is globally watermark-bearing. Besides, to consider potential representation drift caused by advanced attackers, we further introduce a Robust Subspace Registration (RSR) verifier. RSR utilizes sentinel tokens to probe the global manifold distortion. By computing the median deviation of these sentinels as a drift proxy and subtracting it from the observed logits, our method effectively recenters the drifted reasoning manifold back to the canonical signature space.

Our main contributions are summarized as follows: **(1)** We analyze the causal structure of Chain-of-Thought generation and reveal the existence of *structural anchors*. We observe that specific tokens disproportionately govern reasoning dynamics, forming a stable subspace that allows for signal injection without disrupting logical coherence. **(2)** We propose BiCoT, a novel watermarking paradigm that embeds ownership into the geometry of the *reasoning manifold*. Bi-CoT uses bi-level optimization to entangle the watermark with the reasoning process itself, while treating pure control tokens and short-range causal-halo tokens separately. **(3)** We conduct extensive experiments to validate the effectiveness and fidelity of our approach. The results demonstrate that BiCoT achieves strong detection success rates against adaptive attacks while preserving the model's performance on downstream reasoning tasks across benchmark settings.

## 2. Related Work

Existing model watermarking techniques for Large Language Models (LLMs) can be broadly categorized into white-box and black-box paradigms, distinguished primarily by their verification mechanisms and access assumptions.

### 2.1. White-box Watermarking

White-box watermarking operates under the assumption that the verifier has full access to the model's parameters. These approaches typically embed ownership signatures directly into the model weights (Liang et al., 2026). Verification is conducted by inspecting the internal parameters to extract the embedded signal. While these methods allow for robust embedding, the requirement for white-box access significantly limits their utility in scenarios where models are deployed as black-box APIs or encapsulated services.

### 2.2. Black-box Watermarking

In contrast, black-box watermarking verifies ownership solely through API queries, without requiring direct access to the model's internal weights in deployed settings (Yamabe et al., 2025; Xiong et al., 2026). Current literature in this domain is mainly dominated by two primary strategies: backdoor-based methods and model editing-based methods.

**Backdoor-based Approaches.** This stream of research focuses on injecting specific trigger-action patterns into the model (Li et al., 2023; Shao et al., 2025b). The core idea is to force the model to output a specific watermark signal when a predefined trigger appears in the input. However, forcing the model to overfit to a small set of fixed triggers introduces significant drawbacks. It fundamentally compromises *fidelity* on complex tasks and degrades *generalization* due to interference from the backdoor mechanism (Gu et al., 2017; Li et al., 2022; Puah et al., 2024). Additionally, the reliance on fixed patterns renders these watermarks susceptible to detection and filtering, and verification becomes unreliable if triggers are leaked or maliciously spoofed (Wang et al., 2019b; Liu et al., 2019; Guo et al., 2024).

**Model Editing-based Approaches.** To mitigate the inefficiencies of full fine-tuning used in some backdoor methods, model editing-based approaches leverage knowledge updat-

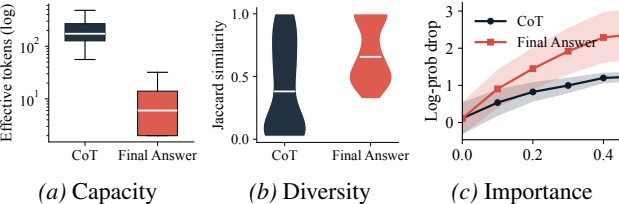

*(a)* Capacity     *(b)* Diversity     *(c)* Importance

*Figure 1.* **CoT provides a richer and less brittle carrier than final answers.** We compare CoT tokens and final-answer tokens with GSM8K-style reasoning traces. (a) *Capacity* is measured by the effective token count, i.e., the exponential of token-level Shannon entropy, where larger values indicate a broader token distribution. (b) *Diversity* is measured by Jaccard similarity between independent generations from the same prompt; lower similarity indicates greater generation diversity. (c) *Importance* is measured by the drop in the log-probability of the correct final answer after randomly masking a given ratio of tokens. CoT exhibits higher capacity, greater diversity, and more distributed importance, while final answers are low-entropy and more sensitive to masking.

ing algorithms to inject watermarks as specific facts (Li et al., 2025b; Yue et al., 2025). While these methods improve efficiency, they treat the watermark as static knowledge to be memorized. Consequently, this paradigm is inherently fragile, relying on rigid question templates that are vulnerable to distribution shifts or adaptive attacks on prompt structures (Cheng et al., 2023; Wang et al., 2024b).

## 3. Revisiting Chain-of-Thought as a Watermark Carrier

This section asks three empirical questions that motivate BiCoT. (1) Is CoT a better carrier than the final answer? (2) Within CoT, are there localized reasoning-sensitive positions that can serve as anchors? (3) Can controlled perturbations to such internal states propagate to output-level signals without disrupting reasoning? The following analysis answers these questions in sequence and motivates Section 4.4.

**Main Settings**. Unless otherwise specified, analyses in this section are conducted on reasoning traces generated by Qwen2.5-7B-Instruct (Yang et al., 2024) on GSM8K (Cobbe et al., 2021). We use correctly solved samples to avoid conflating reasoning failure with representational sensitivity. Hidden states are extracted from the 4th to last transformer layer, and final-answer likelihood is computed under a fixed answer probe. Appendix A provides the full metric definitions, perturbation protocols, and model-/dataset-level consistency checks supporting this.

### 3.1. CoT Is a Richer Carrier than Final Answers

We hereby revisit the assumption that all generated tokens are equally suitable for embedding auxiliary signals. Our analysis is conducted on chain-of-thought (CoT) reasoning traces generated on GSM8K-style mathematical reasoning

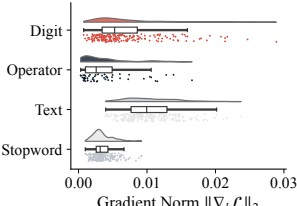

*(a)* Gradient Saliency Distribution

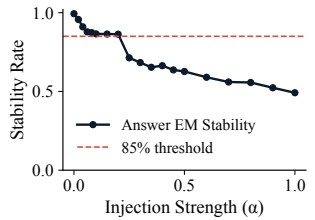

*(b)* Reasoning Utility Resilience under Anchor Perturbation

*Figure 2.* **Structural anchors are identified by causal saliency and validated by perturbation stability.** (a) We compute token saliency as the $\ell_2$ norm of the gradient of the final-answer cross-entropy loss with respect to each CoT hidden state at layer $L - 4$. Digits show a heavy-tailed saliency distribution compared with operators, text tokens, and stopwords, indicating that they carry disproportionate causal influence in mathematical reasoning. (b) We perturb anchor hidden states with increasing injection strength $\alpha$ and measure final-answer exact-match stability. This stable high-saliency regime motivates BiCoT's design: watermark signals are injected into anchor states, while non-anchor tokens are kept orthogonal to preserve the protected LLM's semantic capacity.

tasks. Fig. 1 shows that this assumption does not hold. Across correct generations, CoT exhibits higher effective capacity and substantially greater diversity than the final answer. Moreover, under controlled token masking, importance within CoT is distributed across tokens, whereas the final answer is dominated by a small set of critical tokens and degrades sharply when key positions are masked.

These results indicate that, in structured reasoning settings, CoT forms a larger and more robust representational space than the final answer, making it a more suitable carrier.

### 3.2. Structural Anchors within CoT

Chain-of-thought (CoT) reasoning is often treated as a homogeneous sequence of tokens. However, not all tokens contribute equally to the causal formation of the final answer. We posit that a small subset of tokens acts as *structural anchors*: discrete elements that disproportionately govern reasoning dynamics, possibly in a task-dependent manner.

To identify such anchors, we analyze gradient saliency with respect to hidden states along the reasoning trace. As shown in Fig. 2a, saliency mass is highly concentrated on **digits** in GSM8K-style mathematical reasoning, which exhibit a distinct heavy-tailed distribution. In other domains, the anchor mask is still saliency-defined but may be realized by logical connectives, punctuation, legal predicates, or programming syntax rather than digits. Thus, digits are an instantiation of saliency-defined anchors in mathematical reasoning, rather than a fixed requirement of the BiCoT framework. This separation indicates that the anchor concept is task-dependent rather than lexical, but the underlying geometry is shared.

If these anchors are indeed structural, controlled perturba-

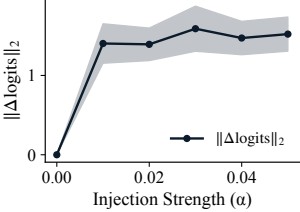
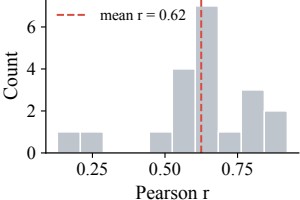

*(a)* Signal propagation from hidden states to output logits.

*(b)* Linearity of signal propagation measured by Pearson $r$.

*Figure 3.* **Controlled CoT perturbations are observable at the output-logit level.** (a) We inject anchor-level perturbations with strength $\alpha$ into CoT hidden states and measure the $\ell_2$ distance between the original and perturbed output logits. The logit response increases approximately linearly in the low-to-moderate injection range and then saturates into a plateau, indicating that latent CoT signals can propagate to observable outputs without abrupt disruption. Shaded regions denote 95% confidence intervals. (b) Pearson correlations between injection strength and output-logit shift across samples. The positive sample-wise correlation distribution suggests a stable pre-saturation propagation regime.

tions should induce a predictable utility–perturbation curve: the model should remain stable under low-to-moderate interventions, while stronger interventions should eventually reduce answer accuracy. Fig. 2b confirms this hypothesis. When perturbations are applied to anchor dimensions, the model maintains high answer stability within a broad low-to-moderate operating regime. Beyond a critical strength, performance eventually degrades smoothly overall and then saturates, revealing a stable pre-saturation linear regime followed by a plateau in the utility–injection tradeoff.

These observations suggest that structural anchors define a low-dimensional subspace where causal influence is both concentrated and resilient. Such a stable regime enables targeted interventions that bind auxiliary signals to the reasoning process while preserving downstream fidelity. This property underlies our subsequent watermarking strategy.

### 3.3. Signal Propagation Through the Reasoning Trace

We analyze signal propagation by injecting controlled perturbations into chain-of-thought (CoT) representations and measuring their effect on the final output in structured reasoning tasks. As shown in Fig. 3, once the injection strength exceeds a small threshold, perturbations in CoT selected anchor states consistently propagate to the output logits.

As the injection strength further increases, the induced output change saturates into a stable plateau. Beyond this point, increasing the strength results in only marginal changes in the output distribution, suggesting that the injected signal has already been fully stably integrated into the internal reasoning process rather than acting as an external bias.

Within the low-to-moderate pre-saturation injection range,

signal propagation exhibits an approximately linear relationship between CoT perturbations and output-logit changes, reflected by stable correlation patterns across samples. This linearity indicates a controllable operating region in which the watermark signal remains effective while preserving downstream task behavior without compromising fidelity.

These observations identify a pre-saturation regime in which watermark signals are simultaneously present, detectable, stable, and non-disruptive. This regime naturally motivates reliable geometric optimization strategies that balance downstream task fidelity with alignment to CoT representations.

## 4. The Proposed Method

### 4.1. Preliminaries

**Threat Model.** We consider a realistic model-misuse scenario involving two parties: the *LLM Provider* and a *Malicious Developer*. The provider owns a source model and releases it under specific license constraints. The malicious developer violates these constraints by deploying a derivative or stolen model for commercial use through an API service. The provider aims to remotely verify whether the suspicious API is derived from the protected source model using limited black-box queries, while the developer aims to evade verification without sacrificing downstream utility.

**Capabilities and Constraints.** We distinguish the watermark embedding stage from the verification stage. During embedding, the provider has white-box access to its own clean source model and can modify its parameters to inject the watermark. During verification, however, the provider has only restricted logit-based black-box access to the suspect API: it can submit ordinary task prompts and observe only the Top-$k$ next-token log-probabilities returned by the API (Li et al., 2025e; Dai et al., 2025). BiCoT does not require access to the suspect model's weights, hidden states, training data, or full-vocabulary logits. The verifier also only uses the returned Top-$k$ log-probabilities to evaluate the watermark signal over observable sentinel tokens.

This assumption corresponds to a practical logprob-enabled black-box API setting. If the suspect service deliberately disables all log-probability outputs and exposes only sampled text, remote distribution-based verification becomes infeasible; we therefore treat strictly text-only APIs as a challenging boundary case outside our current threat model.

To evade detection, the developer may employ adaptive attacks at two levels: (1) *model-level attacks*, such as supervised fine-tuning, parameter-efficient fine-tuning, or common quantization to alter internal signatures; (2) *output-level attacks*, such as perturbing, paraphrasing, or post-processing generated responses. Crucially, all evasion attempts are subject to a strict *utility constraint*: the developer

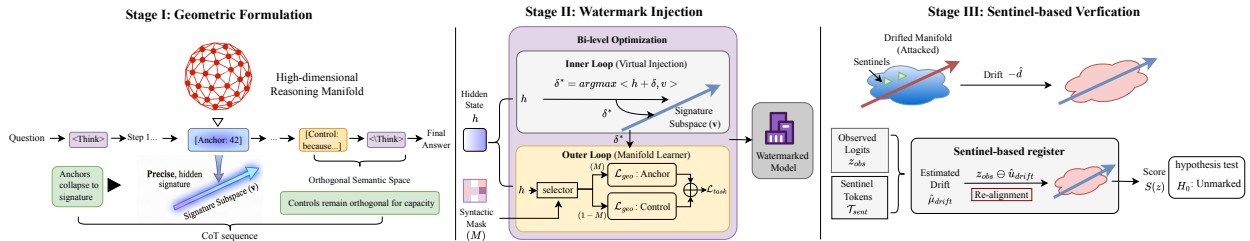

*Figure 4.* **Overview of the BiCoT architecture.** The proposed overall pipeline proceeds in three stages: **Stage I: Geometric Formulation (Left).** We define the reasoning process on a high-dimensional manifold where normalized latent states **h** interact with a compact private signature subspace $\mathcal{S}_{\mathbf{v}}$. Within the CoT sequence, saliency-defined *anchor* tokens are constrained to collapse onto the signature, while pure *control* tokens remain orthogonal to preserve semantic capacity. Tokens immediately following an anchor are treated as a separate *causal-halo* category because they may retain short-range residual alignment (Sec. 4.3). **Stage II: Watermark Injection (Middle).** The injection is formulated as a *Bi-level Optimization* problem. A *Syntactic Mask* ($M(x_t)$) acts as a selector (abbreviated as $M$ in Fig.): the *Inner Loop* computes a virtual hidden-state injection $\delta^*$ for alignment, while the *Outer Loop* jointly updates the model parameters to minimize a combined objective of geometric constraints ($\mathcal{L}_{geo}$) and task utility ($\mathcal{L}_{task}$) (Sec. 4.4). **Stage III: Sentinel-based Verification (Right).** To address distributional drift caused by attacks, we employ a robust sentinel-based register. Using pre-identified sentinel tokens, the system estimates the drift vector and recenters the observed logits $\mathbf{z}_{obs}$ before the final calibrated hypothesis test (Sec. 4.5).

must preserve the model's downstream performance to maintain the practical commercial value of the stolen service.

## 4.2. Overview

We introduce BiCoT, a framework that embeds ownership signatures directly into the geometry of the reasoning manifold. We select saliency-defined anchor tokens as the primary carriers for the watermark, motivated by their strong causal influence on reasoning dynamics. Our analysis shows that in GSM8K-style mathematical reasoning these anchors are predominantly digits, while in other domains they may correspond to logical connectives, punctuation, or syntax tokens. The key design principle is to align anchor states with a secret signature subspace while keeping control tokens orthogonal and treating short-range halo tokens separately.

To ensure that this rigid entanglement does not degrade task performance, we leverage the unused capacity of the linguistic context. As observed in Fig. 1a and 1b, non-anchor tokens (e.g., text connectors) exhibit substantially higher capacity and diversity than the final answer. We utilize these tokens as a flexible *semantic buffer*. In our optimization, while anchors are constrained to collapse, control tokens are encouraged to remain orthogonal to the signature. This allows the model to utilize the high degrees of freedom in the linguistic subspace to compensate for the residual geometric constraints imposed on the digits.

Finally, our bi-level optimization strategy is grounded in the signal propagation analysis of Sec. 3.3. The existence of a linear propagation regime (Fig. 3) implies that latent perturbations on CoT tokens can controllably influence the output distribution without inducing collapse, provided the injection remains within the observed plateau. BiCoT turns the empirical propagation property into a controlled optimization objective in a principled manner, rather than imposing

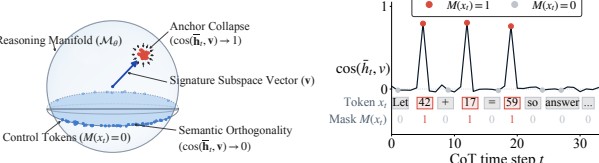

*(a)* Spatial view: manifold collapse. *(b)* Temporal view: trajectory alignment.

*Figure 5.* **Geometry of the reasoning manifold.** (a) On the normalized latent hypersphere, anchor-token representations are encouraged to collapse onto the signature subspace $S_v$, while control-token representations remain orthogonal to preserve semantic capacity. (b) As CoT sequence unfolds, the signature projection $\cos(\bar{h}_t, v)$ remains near zero for control tokens with $M(x_t) = 0$, but spikes at anchor tokens with $M(x_t) = 1$. The token and mask rows illustrate how the watermark is intermittently woven into the reasoning trajectory through structural anchors.

an external output bias. This design enables the watermark signal to remain coupled with reasoning dynamics while avoiding direct interference with answer formation.

## 4.3. Stage I: Geometric Formulation

We model the latent state of a Large Language Model (LLM) $f_\theta$ as a continuous trajectory on a high-dimensional *Reasoning Manifold* $\mathcal{M}_\theta \subset \mathbb{R}^d$ (Li et al., 2025a). Unlike perturbations applied to the discrete output distribution, we define watermarking as an intrinsic geometric constraint on $\mathcal{M}_\theta$. We introduce a proprietary *Signature Subspace* $\mathcal{S}_{\mathbf{v}}$, parameterized by a fixed unit vector $\mathbf{v} \in \mathbb{R}^d$. As visualized in the spatial view of Fig. 5 (a), our method enforces a structural collapse of the manifold onto $\mathcal{S}_{\mathbf{v}}$ for specific anchor states.

The optimization objective forces $\mathcal{M}_\theta$ to align locally with $\mathcal{S}_{\mathbf{v}}$ conditionally based on token semantics. We employ a semantic syntactic indicator $M(x_t) \in \{0, 1\}$ to partition the

sequence into *anchor tokens* and *control tokens*. Fig. 5 (b) illustrates this sequential dynamic: the reasoning trajectory $\mathbf{h}_t$ intermittently aligns with $\mathbf{v}$ when $M(x_t) = 1$ and remains orthogonal otherwise. Formally, we seek a parameter configuration $\theta^*$ satisfying the geometric limit exactly:

$$\lim_{\theta \to \theta^*} \cos(\bar{\mathbf{h}}_t, \mathbf{v}) = \begin{cases} 1 & \text{if } M(x_t) = 1 \quad (collapsed), \\ 0 & \text{if } M(x_t) = 0 \quad (orthogonal). \end{cases} \tag{1}$$

This formulation tightly entangles the watermark with the functional reasoning process, making the signature inseparable from the underlying manifold structure itself.

## 4.4. Stage II: Watermark Injection

We formulate the watermarking process as a bi-level optimization problem. Our objective is to embed a geometric signature $\mathbf{v}$ into the hidden states $\mathbf{h}$ while maintaining task performance. We adopt a bi-level formulation to enforce alignment. In the inner loop, we find a perturbation $\boldsymbol{\delta}^*$ that maximizes the projection onto the signature subspace $\mathcal{S}_{\mathbf{v}}$. In the outer loop, we update the parameters $\theta$ to minimize the task loss and the geometric alignment error. Formally:

$$\min_{\theta} \quad \mathcal{L}_{\text{task}}(f_\theta(\mathbf{h} + \boldsymbol{\delta}^*), y) + \lambda \mathcal{L}_{\text{geo}}(\mathbf{h}, \mathbf{v}) + \gamma \mathcal{L}_{\text{KL}} \tag{2}$$
$$\text{s.t.} \quad \boldsymbol{\delta}^* = \arg\max_{||\boldsymbol{\delta}|| \le \epsilon} \langle \mathbf{h} + \boldsymbol{\delta}, \mathbf{v} \rangle.$$

Here, $\mathcal{L}_{\text{task}}$ represents the cross-entropy loss on the perturbed states, which serves as a robustness regularizer.

The geometric loss $\mathcal{L}_{\text{geo}}$ imposes structural constraints on the reasoning manifold. We introduce a binary mask $M(x_t) \in \{0, 1\}$ to distinguish anchor tokens ($M(x_t) = 1$) from control tokens ($M(x_t) = 0$). The loss is defined as:

$$\mathcal{L}_{geo}(H_\ell, v) = \frac{1}{T} \sum_{t=1}^{T} \left[ M_t \|\bar{h}_{\ell,t} - v\|_2^2 + (1 - M_t)\langle \bar{h}_{\ell,t}, v \rangle^2 \right]. \tag{3}$$

where $\bar{\mathbf{h}}$ denotes the $\ell_2$-normalized hidden state.

Anchors ($M(x_t) = 1$): We minimize the Euclidean distance between the normalized state and the signature $\mathbf{v}$. This term forces the manifold to collapse onto the signature subspace at specific steps. Controls ($M(x_t) = 0$): We minimize the squared cosine similarity. This term enforces orthogonality, preserving the capacity of control tokens for reasoning logic (Sec. 3.1). $\mathcal{L}_{KL}$ denotes the local consistency regularizer between the next-token distributions of the watermarked model and the frozen base model on the same CoT prefix.

**Proposition 4.1** (Equivalence). *For unit vectors $\bar{\mathbf{h}}$ and $\mathbf{v}$, minimizing $\|\bar{\mathbf{h}} - \mathbf{v}\|^2$ is equivalent to maximizing $\cos(\mathbf{h}, \mathbf{v})$. An upper bound on the loss $\xi$ implies a lower bound:*

$$\mathcal{L}_{geo} \le \xi \implies \cos(\mathbf{h}, \mathbf{v}) \ge 1 - \frac{\xi}{2}. \tag{4}$$

This equivalence ensures that by minimizing the geometric loss during training, we explicitly enlarge the cosine margin in the latent space, providing a theoretically guaranteed, stable, and reliable signal-to-noise ratio for verification.

## 4.5. Stage III: Sentinel-based Verification

Sec. 3.3 demonstrated that watermark signals propagate linearly when the model operates within a stable regime. However, real-world attacks (e.g., heavy quantization or SFT) often push the latent representation beyond this linear plateau, inducing a systematic *distributional drift*. This displacement renders simple correlation thresholds unreliable. To address this, we introduce **Robust Subspace Registration (RSR)**, a blind calibration algorithm that adaptively recovers the signature by realigning the drifting manifold.

**Manifold Drift Hypothesis.** We model the observed logits $\mathbf{z}_{obs}$ of a watermarked model under attack as an affine transformation of the clean signature. Let $\mathbf{W}_{head}$ be the projection matrix of the language head. We posit that attacks induce a translation vector $\mathbf{d}_{drift}$ in the hidden space:

$$\mathbf{z}_{obs} = \mathbf{W}_{head}(\mathbf{h} + \mathbf{d}_{drift}) + \boldsymbol{\epsilon} = \mathbf{z}_{clean} + \underbrace{\mathbf{W}_{head}\mathbf{d}_{drift}}_{\text{Systematic Bias}} + \boldsymbol{\epsilon},$$
$$\tag{5}$$

where $\boldsymbol{\epsilon}$ represents unstructured noise. A naive detector fails because the systematic bias strongly dominates the correlation signal, especially under post-deployment attacks.

**Sentinel-Based Registration.** Since we lack access to the ground-truth weights during verification, we employ a set of *Sentinel Tokens* $\mathcal{T}_{sent}$—tokens identified during training as having high sensitivity to the signature subspace. During verification, we compute the robust central tendency (median) of the sentinels' deviation from their pre-computed base statistics to estimate the drift vector $\hat{\boldsymbol{\mu}}_{drift}$. The registered detection score $s$ for a query is then computed by projecting the calibrated logits onto the signature:

$$s(\mathbf{z}_{obs}) = \left\langle \frac{\mathbf{z}_{obs} - \hat{\boldsymbol{\mu}}_{drift} - \boldsymbol{\mu}_{base}}{\|\mathbf{z}_{obs} - \hat{\boldsymbol{\mu}}_{drift} - \boldsymbol{\mu}_{base}\|}, \mathbf{W}_{head}\mathbf{v} \right\rangle. \tag{6}$$

By subtracting $\hat{\boldsymbol{\mu}}_{drift}$, RSR effectively "rotates" the drifted point cloud accurately back to the canonical signature space (as visualized in Fig. 4 *right* VERIFICATION).

**Null-Hypothesis Testing.** The effectiveness of our Z-score aggregation relies on the alignment bound established in Proposition 4.1. We fit a Gaussian null distribution $\mathcal{N}(\mu_0, \sigma_0^2)$ using responses from non-watermarked models for robust calibration under the null. The final verification decision is formulated as a hypothesis test: we reject the null hypothesis $H_0 : \mathbf{h} \perp \mathbf{v}$ if the aggregated Z-score of the registered stream exceeds a confidence level $\alpha$.

**Proposition 4.2** (Drift Invariance). *Assume the estimator $\hat{\boldsymbol{\mu}}_{drift}$ perfectly captures the systematic bias such that*

$\hat{\boldsymbol{\mu}}_{drift} \approx \mathbf{W}_{head}\mathbf{d}_{drift}$. *The RSR score $s(\mathbf{z}_{obs})$ is invariant to any translational manifold shift $\mathbf{d}_{drift}$.*

While Proposition 4.2 ensures signal recovery, real-world verification requires rigorous error control. The following theorem guarantees that RSR prevents false accusations regardless of the attack magnitude, under arbitrary affine perturbations. The proof is provided in Appendix C.1.

**Theorem 4.3** (Robust Type-I Error Control). *Let $\mathcal{H}_0$ be the null hypothesis (non-watermarked model) where logits follow $\mathbf{z} \sim \mathcal{N}(\boldsymbol{\mu}, \mathbf{I})$. Consider an attack that introduces an arbitrary affine shift $\mathbf{d} \in \mathbb{R}^V$. For any significance level $\alpha \in (0,1)$, let $\tau_\alpha$ be the critical threshold derived from the standard normal distribution. The RSR-calibrated test statistic $Z_{RSR}$ satisfies:*

$$\sup_{\mathbf{d} \in \mathbb{R}^V} \mathbb{P}\left(Z_{RSR}(\mathbf{z} + \mathbf{d}) > \tau_\alpha \mid \mathcal{H}_0\right) \leq \alpha. \quad (7)$$

# 5. Experiments

## 5.1. Experimental Setup

**Models.** We conduct all experiments primarily on Qwen2.5-7B-Instruct (Yang et al., 2024), and further evaluate cross-model generalization on Mistral-7B-Instruct (Jiang et al., 2023), DeepSeek-7B-Instruct (Bi et al., 2024), and Qwen2.5-14B-Instruct (Yang et al., 2024). The base model refers to the corresponding clean instruction-tuned checkpoint, serving in practice without any watermark injection.

**Watermarking and Verification.** We embed watermarks using **BiCoT**, which injects a structured signal into the internal reasoning process via bi-level optimization. Watermark detection is performed in a logprob-enabled black-box setting using our RSR verifier, which operates on ordinary task prompts and uses only the returned Top-$k$ next-token log-probabilities for observable sentinel tokens. It does not access model weights, hidden states, gradients, training data, or full-vocabulary logits; strictly text-only or fully hidden-CoT APIs are outside our current threat model.

Unless otherwise stated, we highlight WSR@0.1% FPR as the deployment-critical metric. The baseline iSeal (Xiong et al., 2026), llmmap (Pasquini et al., 2025), IF-sft, IF-emb (Xu et al., 2024a), met (Gao et al., 2025) and SEAL(Dai et al., 2025) are listed in AppendixG for comparison.

**Datasets.** Detection and robustness experiments are conducted on reasoning-oriented prompts in GSM8K (Cobbe et al., 2021) and Alpaca (Taori et al., 2023). Fidelity is evaluated across a comprehensive suite of standard benchmarks, including MMLU (Hendrycks et al., 2021), BoolQ (Clark et al., 2019), Winogrande (Keisuke et al., 2020), WSC (Levesque et al., 2012), PIQA (Bisk et al., 2020), OpenBookQA (Mihaylov et al., 2018), ARC-Challenge (Clark et al., 2018), RTE, MRPC, SST-2, QNLI, QQP, CoLA, MNLI (Wang et al., 2019a), and WiC (Pilehvar & Camacho-Collados, 2019).

## 5.2. Black-box Detectability

Following the settings in (Dai et al., 2025), we benchmark the black-box detectability of BiCoT against state-of-the-art watermarking methods under identical protocols. Tab. 1 summarizes the results on four LLMs. We report detection metrics including Area Under the Curve (AUC), partial AUC (pAUC), Mahalanobis Distance (MD), and Watermark Success Rate (WSR) at low False Positive Rates (FPR).

Our method achieves saturation performance across all models. As shown in Tab. 1, BiCoT consistently attains 1.0 AUC and 100% WSR at both 0.1% and 1.0% FPR. This indicates that the watermark signal injected is robust and distinguishable even under strict detection thresholds.

## 5.3. Fidelity Evaluation

We evaluate the watermarked model on 15 benchmarks covering general reasoning and sentence understanding. As shown in Tab. 2, the watermarked model achieves good utility preservation, with performance comparable to the original baseline. On reasoning tasks, the degradation is marginal ($-0.0083$). On sentence-level tasks, we observe minor fluctuations, ranging from slight gains to small drops.

## 5.4. Ablation Studies

We investigate the effectiveness of individual components in Tab. 3. The results demonstrate that our proposed BiCoT framework achieves optimal detection performance. In the training formulation (Panel A), the watermark loss $L_{wm}$ is foundational; removing it naturally yields random-guess level performance. Crucially, in the detection phase (Panel B), the Re-anchoring mechanism proves vital: ablating this component causes a catastrophic drop in pAUC, validating its necessity for countering distribution drift. Furthermore, while the pruning strategy is not strictly required for high AUC, it significantly boosts the separation margin, ensuring robust detection boundaries in our main experiments.

## 5.5. Sensitivity Analysis

We analyze the sensitivity of our proposed method with respect to three key hyperparameters: the number of sentinels ($N$), the fraction of calibration data, and the noise level ($\sigma$). As shown in Fig. 8, our approach demonstrates strong robustness across all settings. Specifically, the detection performance saturates rapidly with a small number of sentinels (Fig. 8a) and maintains high AUC scores even with limited calibration data (Fig. 8b). Furthermore, the method exhibits negligible performance degradation against noise perturbations (Fig. 8c), confirming its stability in practical

*Table 1.* Quantitative comparison of watermarking detection performance. The arrow (↑) indicates that higher values correspond to better performance. **Ours** demonstrates robust superior performance across all metrics under the following thresholds.

| Model | Metric | Method | | | | | | |
|---|---|---|---|---|---|---|---|---|
| | | iSeal | llmmap | IF-sft | IF-emb | met | SEAL | **OURS** |
| Qwen2.5-7B | AUC ↑ | 0.8416 | 0.3213 | 0.5000 | 0.5000 | 0.7145 | 1.0000 | **1.0000** |
| | pAUC ↑ | 0.5937 | 0.1876 | 0.0000 | 0.0000 | 0.3256 | 1.0000 | **1.0000** |
| | MD ↑ | 1.5486 | 0.7121 | 0.0000 | 0.0000 | 1.2532 | 1.8900 | **11.6750** |
| | WSR @ 0.1% FPR ↑ | 57.58% | 28.81% | 0.00% | 0.00% | 0.00% | 100.00% | **100.00%** |
| | WSR @ 1.0% FPR ↑ | 57.58% | 28.81% | 0.00% | 0.00% | 0.00% | 100.00% | **100.00%** |
| Mistral-7B | AUC ↑ | 0.6028 | 0.0000 | 0.5000 | 0.5000 | 0.8744 | 1.0000 | **1.0000** |
| | pAUC ↑ | 0.1175 | 0.0000 | 0.0000 | 0.0000 | 0.4658 | 1.0000 | **1.0000** |
| | MD ↑ | 0.4196 | 0.1242 | 0.0000 | 0.0000 | 1.6234 | 1.9100 | **10.3839** |
| | WSR @ 0.1% FPR ↑ | 9.09% | 0.00% | 0.00% | 0.00% | 0.65% | 100.00% | **100.00%** |
| | WSR @ 1.0% FPR ↑ | 9.09% | 0.00% | 0.00% | 0.00% | 0.65% | 100.00% | **100.00%** |
| Deepseek-7B | AUC ↑ | 0.9997 | 0.0000 | 0.5000 | 0.5000 | 0.4887 | 1.0000 | **1.0000** |
| | pAUC ↑ | 0.9954 | 0.0000 | 0.0000 | 0.0000 | 0.1768 | 1.0000 | **1.0000** |
| | MD ↑ | 3.8556 | 0.1872 | 0.0000 | 0.0000 | 1.5876 | 1.9000 | **5.4545** |
| | WSR @ 0.1% FPR ↑ | 98.48% | 0.00% | 0.00% | 0.00% | 6.80% | 100.00% | **100.00%** |
| | WSR @ 1.0% FPR ↑ | 98.48% | 0.00% | 0.00% | 0.00% | 23.00% | 100.00% | **100.00%** |
| Qwen2.5-14B | AUC ↑ | 0.9369 | 0.3345 | 0.5000 | 0.5000 | 0.4676 | 1.0000 | **1.0000** |
| | pAUC ↑ | 0.7998 | 0.1946 | 0.0000 | 0.0000 | 0.1176 | 1.0000 | **1.0000** |
| | MD ↑ | 2.2839 | 0.8789 | 0.0000 | 0.0000 | 0.8765 | 1.8900 | **10.2989** |
| | WSR @ 0.1% FPR ↑ | 78.79% | 30.78% | 0.00% | 0.00% | 0.10% | 100.00% | **100.00%** |
| | WSR @ 1.0% FPR ↑ | 78.79% | 30.78% | 0.00% | 0.00% | 0.10% | 100.00% | **100.00%** |

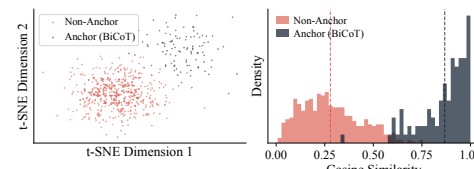

*Figure 6.* **Representation Analysis of Bi-CoT. t-SNE(left):** Separation of anchor tokens. **Hist(right):** Cosine similarities to **v** show clear margin.

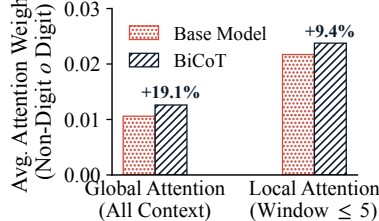

*Figure 7.* **Attention Redistribution.** BiCoT increases the global attention weight significantly.

*Table 2.* **Fidelity Evaluation on Downstream Tasks (Mean ± Std).** We report the accuracy across two categories: General Knowledge (Panel A) and Sentence-level Understanding (Panel B). *Diff* denotes the performance gap ($Acc_{water} - Acc_{orig}$).

| Task | Original | BiCoT-Watermarked | Diff ($\Delta$) |
|---|---|---|---|
| *Panel A: General Knowledge* | | | |
| MMLU | $0.7320 \pm 0.0000$ | $0.7237 \pm 0.0000$ | $-0.0083 \pm 0.0000 \downarrow$ |
| BoolQ | $0.8344 \pm 0.3717$ | $0.8581 \pm 0.3490$ | $+0.0237 \pm 0.0000 \uparrow$ |
| Winogrande | $0.8150 \pm 0.3883$ | $0.8095 \pm 0.3927$ | $-0.0055 \pm 0.0000 \downarrow$ |
| WSC | $0.4386 \pm 0.4962$ | $0.4657 \pm 0.4988$ | $+0.0271 \pm 0.3361 \uparrow$ |
| PIQA | $0.8580 \pm 0.3491$ | $0.8575 \pm 0.3496$ | $-0.0005 \pm 0.2523 \downarrow$ |
| OpenBookQA | $0.8680 \pm 0.3385$ | $0.8520 \pm 0.3551$ | $-0.0160 \pm 0.2603 \downarrow$ |
| ARC-Challenge | $0.9010 \pm 0.2986$ | $0.8976 \pm 0.3032$ | $-0.0034 \pm 0.1703 \downarrow$ |
| *Panel B: Sentence-level Understanding* | | | |
| RTE | $0.8592 \pm 0.3478$ | $0.8917 \pm 0.3108$ | $+0.0325 \pm 0.2599 \uparrow$ |
| MRPC | $0.7108 \pm 0.4534$ | $0.7721 \pm 0.4195$ | $+0.0613 \pm 0.3411 \uparrow$ |
| SST-2 | $0.9243 \pm 0.2645$ | $0.9358 \pm 0.2451$ | $+0.0115 \pm 0.1432 \uparrow$ |
| QNLI | $0.8574 \pm 0.3497$ | $0.8666 \pm 0.3401$ | $+0.0092 \pm 0.2684 \uparrow$ |
| QQP | $0.7833 \pm 0.4120$ | $0.8223 \pm 0.3822$ | $+0.0390 \pm 0.2566 \uparrow$ |
| CoLA | $0.7699 \pm 0.4209$ | $0.7162 \pm 0.4508$ | $-0.0537 \pm 0.3112 \downarrow$ |
| MNLI | $0.8292 \pm 0.3763$ | $0.8579 \pm 0.3492$ | $+0.0286 \pm 0.2638 \uparrow$ |
| WiC | $0.6097 \pm 0.4878$ | $0.6317 \pm 0.4824$ | $+0.0219 \pm 0.3257 \uparrow$ |

*Table 3.* **Comprehensive Ablation Study.** We investigate the contribution of individual components in both the training (Panel A) and detection (Panel B) phases under controlled settings. The "Full" setting denotes our proposed BiCoT framework. All models in Panel A are evaluated against an unwatermarked SFT baseline.

| Setting | AUC | pAUC | MD |
|---|---|---|---|
| *Panel A: Training Formulation* | | | |
| **Full Objective** | **1.0000** | **1.0000** | **11.64** |
| w/o Watermark Loss ($L_{wm}$) | 0.1187 | 0.0000 | 1.69 |
| w/o Bi-level Optimization | 0.9986 | 0.9803 | 4.72 |
| *Panel B: Detection Inference* | | | |
| **Full Detector (RSR)** | **1.0000** | **1.0000** | **11.64** |
| w/o Re-anchoring | 0.8770 | 0.4100 | 1.70 |
| w/o Global Aggregation | 0.9782 | 0.8278 | 2.79 |
| w/o Pruning (MAD) | 1.0000 | 1.0000 | 10.01 |

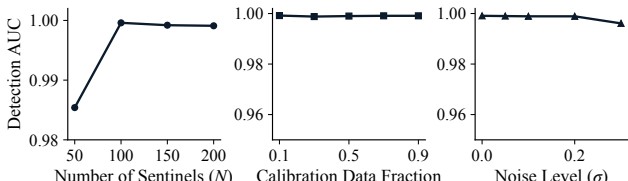

*Figure 8.* **Sensitivity Analysis.** We evaluate the robustness of the Detection AUC under three distinct settings: (a) Sentinel Efficiency: The performance consistently improves with the number of sentinels ($N$) and saturates when $N > 100$, indicating that a small sentinel set is sufficient. (b) Data Efficiency: The method achieves competitive, near-saturated AUC even with a minimal fraction (0.1) of calibration data. (c) Noise Robustness: The detection capability is resilient to noise injection, maintaining high accuracy across all ranges even as the noise level $\sigma$ increases.

set, making the detection procedure lightweight. The rapid saturation with respect to $N$ also indicates that the selected sentinels capture the dominant signature-bearing directions in the output space. Meanwhile, the flat response under increasing $\sigma$ shows that RSR more reliably and consistently absorbs random perturbations rather than mistaking them for watermark evidence under deployment noise.

## 5.6. Robustness Against Attacks

We further evaluate the robustness of BiCoT under a range of model-level and output-level attacks. Tab. 4 summarizes the detection metrics. We observe that the proposed method is highly robust to common perturbations. Specifically, under Gaussian noise ($\sigma = 0.05$) and full-parameter Supervised Fine-Tuning (SFT), the method maintains near-perfect detection performance (AUC ≈ 1.0) and retains a watermark success rate (WSR) of 100% at 1% FPR. Fur-

scenarios. These trends suggest that the verifier does not rely on an excessively large sentinel pool or dense calibration

*Table 4.* **Robustness against various attacks.** We report the detection performance (AUC, pAUC, Mahalanobis Distance, and WSR at low FPRs) of the watermarked model. The evaluation is split into two categories: Common Perturbations (top) including noise and supervised fine-tuning (SFT), and Advanced Attacks (bottom) including PEFT, quantization (QT), and ANCHOR attacks. The baseline (Random Key) yields $\approx 0.5$ AUC across all settings.

| Metric | No Attack | Noise ($\sigma$=0.05) | SFT (Alpaca) | SFT (GSM8K) |
|---|---|---|---|---|
| AUC | 1.0000 | 1.0000 | 1.0000 | 1.0000 |
| pAUC | 1.0000 | 1.0000 | 1.0000 | 1.0000 |
| MD | 11.675 | 5.0628 | 4.9495 | 8.4026 |
| WSR @0.1% FPR | 100% | 100% | 99.55% | 100% |
| WSR @1.0% FPR | 100% | 100% | 100% | 100% |

| Metric | PEFT (Alpaca) | PEFT (GSM8K) | MM | QT | ANCHOR |
|---|---|---|---|---|---|
| AUC | 1.0000 | 1.0000 | 0.9994 | 0.9896 | 0.9998 |
| pAUC | 1.0000 | 1.0000 | 0.9970 | 0.9688 | 0.9954 |
| MD | 6.7597 | 6.9442 | 4.7927 | 3.6463 | 3.5764 |
| WSR @0.1% FPR | 100% | 100% | 98.33% | 94.55% | 98.48% |
| WSR @1.0% FPR | 100% | 100% | 99.70% | 96.52% | 98.48% |

thermore, the robustness generalizes to parameter-efficient tuning and compression scenarios. As shown in the bottom panel of Tab. 4, the watermark survives PEFT (LoRA) without degradation and maintains high stability even under aggressive quantization (QT) and other attack methods.

Of particular note is the resilience against ANCHOR, a targeted adaptive attack where the adversary explicitly masks all digit tokens during inference to bypass detection. Remarkably, our method remains impervious even under masking to this rigorous setting. We attribute this to the intrinsic redistribution of the model's attention mechanism: as analyzed in Fig. 7. Detailed attack configurations and their mapping to the threat model are provided in Appendix E.1.

### 5.7. Effectiveness Analysis

**Manifold Separation.** As shown in Fig. 6 (left), BiCoT induces a clear structural shift in the representation space. The watermarked representations form a compact subspace distinct from the diverse manifold of natural reasoning, confirming that the watermark is semantically preserved yet geometrically isolatable. The t-SNE visualization further shows that anchor states are clustered more tightly after watermarking, while non-anchor states remain broadly dispersed. This separation suggests that BiCoT selectively introduces a localized geometric signature rather than globally collapsing the entire reasoning manifold in practice.

**Detection Robustness.** Fig. 6 (right) further demonstrates that the two distributions are statistically disjoint with clear, robust, wide margins. Natural traces exhibit near-zero cosine similarity (orthogonality in high dimensions), while watermarked traces align perfectly with the secret key.

**Attention Redistribution.** We further investigate the internal mechanism by analyzing the attention flow from non-anchor tokens to anchor tokens. As shown in Fig. 7, BiCoT

exhibits a substantial increase in global attention towards numerical or syntactic anchor tokens compared to the base model. Notably, this global shift outweighs purely local attention boosts, indicating that the model learns to aggregate anchor information across the long-range contexts.

**Causal-Halo Locality.** Although Eq. 3 regularizes control tokens to be orthogonal to the signature direction, autoregressive generation may create a transient "causal halo" on tokens immediately following an anchor. This residual signal rapidly decays, and pure text tokens remain nearly orthogonal, indicating that the watermark is localized around high-saliency anchor states rather than globally injected into the reasoning trace. We provide token-level and corpus-level evidence across different token positions in Appendix F.

### 5.8. Threat Model Boundary and Limitation: Logprob Enabled Verification

BiCoT assumes a logprob-enabled black-box setting, where the verifier can access Top-$k$ next-token log-probabilities for a small set of sentinel tokens. This is a deliberate boundary rather than an implementation artifact: detecting ownership through likelihood patterns around high-saliency reasoning anchors allows BiCoT to avoid perturbing final answers or injecting brittle textual triggers into the chain of thought.

This assumption is also practical for many modern LLM APIs and open-source serving engines, where token-level likelihoods support constrained decoding, structured generation, confidence estimation, and evaluation. Although a malicious host could disable such outputs, doing so would reduce compatibility with common downstream toolchains and remove functionality expected by many applications, creating a non-trivial economic cost of circumvention.

We nevertheless acknowledge that the Top-$k$ budget affects verification in deployed systems. Our current detector uses an effective setting of $k = 40$. This motivates further study of low-$k$ verification, including more refined sentinel-token selection, query aggregation, and threshold calibration under truncated probability observations. We provide a detailed survey in Appendix E.

## 6. Conclusion

We presented BiCoT, a watermarking framework that embeds ownership signals into CoT reasoning representations rather than final answers. By aligning salient anchors with a signature subspace while keeping control tokens orthogonal, BiCoT ties the watermark to reasoning with minimal utility loss. RSR further controls representation drift after real-world deployment. Experiments across models, datasets, and attacks demonstrate robust black-box detectability and negligible fidelity loss, showing CoT representations as a practical and reliable space for LLM ownership verification.

## Impact Statement

This work contributes to trustworthy AI by improving ownership verification for large language models. BiCoT helps detect unauthorized model copying and unlicensed commercial deployment while preserving downstream reasoning performance. By embedding watermarks into the CoT reasoning process rather than modifying the final answers, the method may support model provenance, intellectual property protection, and responsible model release. However, the watermarking systems should be used carefully: verification results should be interpreted with proper false-positive control, privacy-preserving query protocols, and independent auditing, especially for the legal or commercial claims.

## Acknowledgments

This work was supported in part by the National Natural Science Foundation of China (NO. 62472284), Shanghai Key Laboratory of Scalable Computing and Systems, SJTU Kunpeng & Ascend Center of Excellence, and Shanghai Jiao Tong University under "Bo Le" Grant AO120001/074.

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

## A. Supplementary Analysis for Chain-of-Thought Token Roles

This appendix supplements Section 3.1 with the formal metric definitions, perturbation protocols, and experimental configurations used to motivate BiCoT.

### A.1. Experimental Setup

Let $x$ denote the input prompt, $c = (c_1, \ldots, c_T)$ denote the generated chain-of-thought tokens, and $y$ denote the ground-truth final answer. For each correctly solved sample, we evaluate the likelihood of $y$ conditioned on the prompt and the observed reasoning trace. When perturbing or masking a token span $S$, we use a fixed answer probe $q$ and compute the answer likelihood as

$$\ell(S) = \log P_\theta(y \mid x \oplus S \oplus q), \tag{8}$$

where $\oplus$ denotes sequence concatenation. This fixed-probe design ensures that changes in $\ell(S)$ reflect the contribution of the inspected span rather than differences in generated answer format.

For hidden-state analyses, we use the representation $h_t^{L-4}$ of token $t$ at layer $L-4$. This layer is close enough to the output distribution to expose reasoning-sensitive features, while still being internal enough to capture latent chain-of-thought structure before final token selection.

### A.2. Metrics for Token Role Analysis

We compare Chain-of-Thought (CoT) tokens and final-answer tokens using three complementary metrics: effective capacity, diversity, and token importance.

**Effective Capacity.** We quantify the information density of a generated span using the effective token count. For a span $S$ with empirical token distribution $\hat{p}(v)$ over vocabulary $\mathcal{V}$, capacity is defined as

$$C(S) = \exp\left(-\sum_{v \in \mathcal{V}} \hat{p}(v) \log \hat{p}(v)\right). \tag{9}$$

A larger value indicates a broader token distribution and thus a higher-capacity carrier for auxiliary signals.

**Diversity.** We measure generation diversity by sampling two independent outputs $S_A$ and $S_B$ from the same prompt and computing their Jaccard similarity:

$$J(S_A, S_B) = \frac{|S_A \cap S_B|}{|S_A \cup S_B|}. \tag{10}$$

Here each span is treated as a bag of unique unigrams. Lower Jaccard similarity indicates greater diversity. In our analysis, CoT spans exhibit lower similarity than final-answer spans, suggesting that reasoning traces provide a less collapsed and more variable representational space.

**Token Importance.** To measure how much a token or span contributes to recovering the correct answer, we compute the likelihood drop caused by masking or removing that content. For random masking at ratio $r$, let $\tilde{S}_r$ denote the span after masking a fraction $r$ of its tokens. The span-level importance is

$$\Delta\mathcal{L}(r; S) = \log P_\theta(y \mid x \oplus S \oplus q) - \log P_\theta(y \mid x \oplus \tilde{S}_r \oplus q). \tag{11}$$

A gradual increase in $\Delta\mathcal{L}$ indicates distributed importance, while a sharp increase indicates that the target answer relies on a small number of fragile tokens.

For token-level analysis, let $S_{\setminus i}$ denote the span after removing token $s_i$. We define the individual importance of token $s_i$ as

$$\Delta\mathcal{L}_i = \log P_\theta(y \mid x \oplus S \oplus q) - \log P_\theta(y \mid x \oplus S_{\setminus i} \oplus q). \tag{12}$$

A token is considered *critical* if $\Delta\mathcal{L}_i > \eta$, where $\eta$ is a fixed significance threshold shared across spans, models, and datasets. The critical-token rate is then

$$\text{Critical}(S) = \frac{1}{|S|} \sum_{i=1}^{|S|} \mathbb{I}\left[\Delta\mathcal{L}_i > \eta\right]. \tag{13}$$

*Table 5.* **Model- and dataset-level consistency of token-role asymmetry.** Across 12 model-dataset combinations, final-answer spans consistently exhibit larger likelihood drops and substantially higher critical-token rates than CoT spans. This supports the claim that CoT provides a more distributed and less brittle carrier, while final answers concentrate answer-critical evidence in a small number of tokens.

| Model | Dataset | CoT Max $\Delta\mathcal{L}$ | Answer Max $\Delta\mathcal{L}$ | CoT Critical % | Answer Critical % |
|---|---|---|---|---|---|
| Qwen2.5-7B | GSM8K | 0.357 | 1.784 | 0.5 | 23.1 |
| Qwen2.5-7B | SVAMP | 0.060 | 2.586 | 0.1 | 28.3 |
| Qwen2.5-7B | CommonsenseQA | 0.047 | 1.055 | 0.1 | 13.8 |
| Qwen2.5-7B | ARC-Challenge | 0.074 | 3.522 | 0.3 | 18.3 |
| Mistral-7B | GSM8K | 0.477 | 2.724 | 0.3 | 34.2 |
| Mistral-7B | SVAMP | 0.200 | 3.080 | 0.4 | 31.4 |
| Mistral-7B | CommonsenseQA | 0.212 | 0.951 | 0.5 | 19.6 |
| Mistral-7B | ARC-Challenge | 0.322 | 2.619 | 0.6 | 22.3 |
| Gemma-2-9B | GSM8K | 0.104 | 2.271 | 0.0 | 29.6 |
| Gemma-2-9B | SVAMP | 0.037 | 2.435 | 0.0 | 29.5 |
| Gemma-2-9B | CommonsenseQA | 0.010 | 1.475 | 0.0 | 30.1 |
| Gemma-2-9B | ARC-Challenge | 0.045 | 3.716 | 0.0 | 44.0 |
| All models | Overall | 0.134 | 2.312 | 0.2 | 26.5 |

### A.3. Model- and Dataset-Level Consistency Checks

To verify that the token-role observations are not specific to Qwen2.5-7B on GSM8K, we repeat the token-importance analysis across multiple model and dataset combinations. For each combination, we evaluate 1,000 samples using the same fixed-probe protocol. Tab. 5 reports the maximum likelihood drop and critical-token rate for CoT and final-answer spans.

The overall trend is stable across models and tasks. On average, the maximum likelihood drop is 2.312 for final-answer spans but only 0.134 for CoT spans. Similarly, the critical-token rate is 26.5% for final-answer spans but only 0.2% for CoT spans. These results indicate that the final answer is a brittle payload: a small number of tokens dominate answer recovery. In contrast, CoT spreads answer-relevant information over a larger reasoning context, making it a more suitable carrier for watermark signals.

### A.4. Identification of Structural Anchors

Section 3.1 further shows that not all CoT tokens contribute equally to the final answer. We identify *structural anchors* as CoT tokens whose hidden states exert disproportionate causal influence on final-answer likelihood.

**Gradient Saliency.** For each CoT token $c_t$, we compute the gradient of the final-answer cross-entropy loss with respect to its hidden state at layer $L - 4$. The saliency score is

$$S_t = \left\| \nabla_{h_t^{L-4}} \mathcal{L}_{\text{CE}}(y \mid x, c) \right\|_2. \tag{14}$$

Large $S_t$ means that small perturbations to $h_t^{L-4}$ have a strong effect on the model's ability to recover the correct final answer.

In GSM8K-style mathematical reasoning, numerical tokens exhibit a heavy-tailed saliency distribution compared with operators, ordinary text tokens, and stopwords. Therefore, our primary experiments instantiate structural anchors using digit tokens. This choice is not hard-coded into the framework: in other domains, the anchor mask can be defined by the same saliency criterion and may correspond to logical connectives, legal predicates, programming syntax, or other task-specific reasoning markers.

*Table 6.* **Sensitivity to the reliability threshold $\tau$.** Across thresholds $\tau \in \{0.75, 0.80, 0.85, 0.90, 0.95\}$, retained correctness remains high and the admissible perturbation interval remains unchanged. This suggests that the conclusion relies on a broad stable plateau rather than on the specific choice of $\tau = 0.85$.

| Thresholds | Avg. retained correctness | Worst-seed range | Admissible interval |
|---|---|---|---|
| $\{0.75, 0.80, 0.85, 0.90, 0.95\}$ | 0.975–0.986 | 0.959–1.000 | $[0, 6]$ |

## A.5. Perturbation Stability and Threshold Sensitivity

To test whether structural anchors provide a stable operating regime for watermark insertion, we perturb anchor hidden states and measure retained final-answer correctness. For an anchor token $c_t$, we construct

$$\tilde{h}_t^{L-4} = h_t^{L-4} + \alpha \mathbf{u}, \tag{15}$$

where $\alpha$ is the perturbation strength and $\mathbf{u}$ is a unit-norm perturbation direction. We then continue the forward pass using $\tilde{h}_t^{L-4}$ and compute exact-match correctness of the final answer.

The retained correctness at strength $\alpha$ is

$$R(\alpha) = \mathbb{E}_{(x,y)} \left[ \mathbb{I} \left( \text{EM}(f_\theta(x; \alpha), y) = 1 \right) \right], \tag{16}$$

where the expectation is taken over originally correct samples. We define the stable operating regime under reliability threshold $\tau$ as

$$\mathcal{I}_\tau = \{\alpha : R(\alpha) \geq \tau\} . \tag{17}$$

The main text uses $\tau = 0.85$ as an operational reliability threshold for non-destructive perturbations. This value is not intended to be uniquely optimal. Rather, it marks a conservative region in which anchor perturbations remain compatible with downstream reasoning utility. To verify that the conclusion does not depend on this specific cutoff, we evaluate thresholds $\tau \in \{0.75, 0.80, 0.85, 0.90, 0.95\}$ on GSM8K and Alpaca using 1,000 test examples and 5 noise seeds. As shown in Tab. 6, the retained correctness remains high across all tested thresholds.

These results show that structural anchors are both causally salient and perturbation-tolerant in a low-to-moderate intervention range. This combination is crucial for BiCoT: the watermark can be coupled to reasoning-relevant states without abruptly destroying the final-answer behavior.

## A.6. Signal Propagation Dynamics

Finally, we analyze whether perturbations to internal CoT representations are observable from output-level signals. This is important because BiCoT verification is ultimately performed through model outputs rather than direct hidden-state access.

We inject additive interference into CoT hidden states at layer $L - 4$:

$$\tilde{h}_t^{L-4} = h_t^{L-4} + \alpha \mathbf{v}, \tag{18}$$

where $\mathbf{v}$ is a unit-norm direction and $\alpha$ controls the injection strength. We then measure the induced shift in output logits:

$$\delta(\alpha) = \left\| \text{Logits}(\tilde{h}^{L-4}) - \text{Logits}(h^{L-4}) \right\|_2 . \tag{19}$$

To quantify whether this propagation is controllable, we compute the Pearson correlation between injection strength $\alpha$ and logit shift $\delta(\alpha)$ over the probing range:

$$\rho = \text{Corr}(\alpha, \delta(\alpha)) . \tag{20}$$

In the low-to-moderate injection regime, the logit response increases approximately linearly with $\alpha$ before saturating into a plateau. This indicates that latent CoT perturbations can propagate to observable output distributions in a stable and non-disruptive manner. The resulting pre-saturation regime motivates BiCoT's optimization design: watermark signals are aligned with reasoning-sensitive anchor states while task fidelity and non-anchor orthogonality constrain the intervention to remain behavior-preserving.

# B. Additional Method Details

This section provides implementation-level details for the BiCoT training objective and discusses how the anchor-token construction generalizes beyond the arithmetic setting.

## B.1. Composite Training Objective

To ensure the watermark is inseparable from reasoning without degrading generation quality, we optimize a composite objective function $\mathcal{L}_{total}$:

$$\mathcal{L}_{total} = \mathcal{L}_{task} + \lambda_1 \mathcal{L}_{WM} + \lambda_2 \mathcal{L}_{Ortho} + \lambda_3 \mathcal{L}_{KL} \tag{21}$$

- $\mathcal{L}_{WM}$ (Watermark Loss): An MSE loss minimizing $||\hat{h}_{anchor} - v||^2$, forcing anchor representations to align with the signature.

- $\mathcal{L}_{Ortho}$ (Orthogonal Loss): Explicitly minimizes the cosine similarity between *control* (non-anchor) tokens and $v$. This constraint prevents watermark "bleed" into normal text, ensuring that general text quality remains unaffected.

- $\mathcal{L}_{KL}$ (Consistency Loss): Constrains the logical divergence between the watermarked model and the base model, preserving the integrity of the Chain-of-Thought reasoning path.

## B.2. Generalization of Anchors and Attack Resilience

**Token-Agnostic Anchoring.** While our primary experiments on GSM8K utilize Arabic numerals as anchors to align with the mathematical reasoning task, the BiCoT framework is token-agnostic. The "Anchor" is simply a logical mask $M_{anchor}$ applied during training. For broader domains, this generalizes to:

- **Syntactic Anchors:** Leveraging specific Part-of-Speech (POS) tags (e.g., verbs in imperative instructions).

- **Symbolic Anchors:** Targeting logical connectors (e.g., "implies", "therefore") in symbolic reasoning tasks.

**Robustness Against Masking Attacks.** The "ANCHOR attack" (physically masking anchor tokens such as digits during inference) fails to evade detection because the watermark is injected into the *hidden states* $h$ preceding the final token selection. Even if the anchor token itself is suppressed via `bad_words_ids`, the rotation of $h$ towards the secret vector $v$ persists. This rotation influences the probability distribution of the *sentinel tokens* (high-frequency neutral words). The sentinels effectively act as a distributed sensor array, detecting the presence of the watermark in the latent space regardless of the final token emitted.

# C. Theoretical Guarantees for Robust Subspace Registration

This section collects the formal guarantees for the Robust Subspace Registration (RSR) verifier. We first prove the drift-invariance property used by the verification score, then prove the Type-I error control result, and finally discuss how the implementation handles non-affine distortions in practice.

## C.1. Proof for Proposition 4.2

*Proof.* Substituting $\mathbf{z}_{obs}$ into Eq. (6), the term $\hat{\boldsymbol{\mu}}_{drift}$ cancels the attack-induced bias $\mathbf{W}_{head}\mathbf{d}_{drift}$. The score reduces to $\langle \mathbf{z}_{clean} + \boldsymbol{\epsilon}, \mathbf{W}_{head}\mathbf{v} \rangle$, which depends solely on the clean alignment and random noise, independent of the drift magnitude $||\mathbf{d}_{drift}||$. □

## C.2. Proof of Theorem 4.3: Robust Type-I Error Control

In this section, we provide a formal derivation for **Theorem 3**. We establish that the Robust Subspace Registration (RSR) algorithm induces a test statistic that is *location-invariant* with respect to the underlying manifold drift. Consequently, we prove that the False Positive Rate (FPR) is strictly bounded by the significance level $\alpha$, regardless of the magnitude of the attack vector $||\mathbf{d}||$.

C.2.1. FORMAL SETUP AND DEFINITIONS

Let the latent space be $\mathcal{X} \subseteq \mathbb{R}^V$. We consider a query sample $\mathbf{z} \in \mathcal{X}$ and a set of sentinel tokens $\mathcal{S} = \{\mathbf{s}_1, \ldots, \mathbf{s}_K\} \subset \mathcal{X}$.

**Hypothesis Definition.**

- **Null Hypothesis ($\mathcal{H}_0$):** The model is unwatermarked. The clean logits for both query and sentinels are independent and identically distributed (i.i.d.) drawn from a centered isotropic Gaussian background process:

$$\mathbf{z}_{clean}, \mathbf{s}_{k,clean} \sim \mathcal{N}(\mathbf{0}, \sigma^2 \mathbf{I}). \tag{22}$$

- **Drift Attack Model:** An adversary applies an affine transformation to the manifold. For any input, the observed logits are displaced by a constant drift vector $\mathbf{d} \in \mathbb{R}^V$:

$$\mathbf{z}_{obs} = \mathbf{z}_{clean} + \mathbf{d}, \quad \mathbf{s}_{k,obs} = \mathbf{s}_{k,clean} + \mathbf{d}. \tag{23}$$

**Estimator Definition.** Let $\hat{\boldsymbol{\mu}} : \mathbb{R}^{K \times V} \to \mathbb{R}^V$ be the robust location estimator. We employ the dimension-wise median operator over the sentinel set $\mathcal{S}_{obs}$. For the $j$-th dimension:

$$\hat{\mu}_j(\mathcal{S}_{obs}) = \text{median}(\{s_{k,obs}^{(j)}\}_{k=1}^K). \tag{24}$$

C.2.2. INVARIANCE LEMMA

We first establish the translation equivariance property of the estimator and the resulting invariance of the calibrated representation.

**Lemma C.1** (Affine Equivariance of the Median). *For any drift vector $\mathbf{d} \in \mathbb{R}^V$, the estimator satisfies $\hat{\boldsymbol{\mu}}(\mathcal{S}_{obs}) = \hat{\boldsymbol{\mu}}(\mathcal{S}_{clean}) + \mathbf{d}$.*

*Proof.* Consider the $j$-th dimension. The observed set is $\{s_{k,clean}^{(j)} + d_j\}_{k=1}^K$. Since the median is a location-equivariant statistic (i.e., $\text{med}(X + c) = \text{med}(X) + c$ for scalar $c$), we have:

$$\hat{\mu}_j(\mathcal{S}_{obs}) = \text{median}(\{s_{k,clean}^{(j)} + d_j\}) = \text{median}(\{s_{k,clean}^{(j)}\}) + d_j = \hat{\mu}_j(\mathcal{S}_{clean}) + d_j. \tag{25}$$

This holds for all dimensions $j \in \{1, \ldots, V\}$, proving the vector equality. □

C.2.3. PROOF OF ERROR CONTROL

**Theorem 3 Statement.** *Under $\mathcal{H}_0$, for any significance level $\alpha \in (0, 1)$ and corresponding critical value $\tau_\alpha = \Phi^{-1}(1 - \alpha)$, the RSR test statistic $Z_{RSR}$ satisfies:*

$$\sup_{\mathbf{d} \in \mathbb{R}^V} \mathbb{P}(Z_{RSR}(\mathbf{z}_{obs}) > \tau_\alpha \mid \mathcal{H}_0) \leq \alpha. \tag{26}$$

*Proof.* The RSR test statistic is defined as the projection of the registered query onto the signature unit vector $\mathbf{v}$:

$$S(\mathbf{z}_{obs}) = \langle \mathbf{z}_{obs} - \hat{\boldsymbol{\mu}}(\mathcal{S}_{obs}), \mathbf{v} \rangle. \tag{27}$$

Substituting the drift model and applying Lemma C.1:

$$\begin{aligned} S(\mathbf{z}_{obs}) &= \langle (\mathbf{z}_{clean} + \mathbf{d}) - (\hat{\boldsymbol{\mu}}(\mathcal{S}_{clean}) + \mathbf{d}), \mathbf{v} \rangle \\ &= \langle \mathbf{z}_{clean} - \hat{\boldsymbol{\mu}}(\mathcal{S}_{clean}), \mathbf{v} \rangle \\ &= S(\mathbf{z}_{clean}). \end{aligned} \tag{28}$$

Equation (28) reveals that the statistic depends *exclusively* on the clean distribution $\mathcal{N}(\mathbf{0}, \sigma^2 \mathbf{I})$, and is algebraically independent of $\mathbf{d}$.

We now analyze the distribution of $S(\mathbf{z}_{clean})$. Let $\mathbf{r} = \mathbf{z}_{clean} - \hat{\boldsymbol{\mu}}(\mathcal{S}_{clean})$. Since $\mathbf{z}_{clean}$ and $\mathcal{S}_{clean}$ are drawn from centered Gaussians, $\mathbf{r}$ is a zero-mean random vector. The projection of a Gaussian vector onto a fixed direction $\mathbf{v}$ remains Gaussian. Therefore, $S(\mathbf{z}_{clean}) \sim \mathcal{N}(0, \sigma_{eff}^2)$.

The standardized Z-score is given by $Z_{RSR} = S(\mathbf{z}_{clean})/\sigma_{eff}$. By definition, $Z_{RSR} \sim \mathcal{N}(0,1)$. The probability of observing a False Positive is given by the tail mass of the standard normal distribution:

$$\mathbb{P}(Z_{RSR} > \tau_\alpha \mid \mathcal{H}_0) = 1 - \Phi(\tau_\alpha) = \alpha. \tag{29}$$

Since Eq. (28) holds for *any* $\mathbf{d}$, the supremum of this probability over the entire vector space $\mathbb{R}^V$ is exactly $\alpha$. The RSR detector is thus uniformly robust against affine drift attacks. □

### C.3. Robustness Mechanism: Handling Non-Affine Drift

**RSR and MAD Pruning.** A critical concern is whether the Robust Subspace Registration (RSR) can handle non-linear distortions (e.g., layer-selective drift or rotational shifts induced by quantization). Our implementation addresses this through a two-stage signal purification process:

1. **Global Drift Correction:** We compute the median shift vector $\Delta = \text{median}(L_{obs}) - L_{base}$ to neutralize global affine translations caused by model fine-tuning or quantization.

2. **Dynamic Subspace Pruning:** To handle non-affine or high-variance noise, we employ Median Absolute Deviation (MAD) filtering. We calculate the $z$-score of the drift for each sentinel dimension $j$:

$$z_j = 0.6745 \cdot \frac{|d_j - \text{median}(d)|}{\text{MAD}(d)} \tag{30}$$

Dimensions where $z_j > 3.5$ are identified as "unhealthy" (i.e., exhibiting non-linear distortion) and are masked out. This ensures that verification relies exclusively on the subspace of the vocabulary that preserves the linear characteristics of the watermark, ensuring robustness against localized rotational drift.

## D. Experimental Protocol and Attack Implementation

This section consolidates the experimental settings that are used throughout Section 5, including the training hyperparameters, the black-box verification protocol, and the attack implementations used for robustness evaluation.

### D.1. Training, Watermarking, and Verification Hyperparameters

We summarize the specific hyperparameters and configuration settings used in our experiments in Tables 7, 8, and 9.

*Table 7.* Model and Training Hyperparameters

| Parameter | Value |
|---|---|
| Base Models | Qwen-2.5-Instruct (7B, 14B) |
| Precision | `bfloat16` (with Flash Attention 2) |
| Optimizer | `paged_adamw_32bit` |
| Learning Rate | $2 \times 10^{-4}$ |
| Batch Size | 2 per device (Effective 16 w/ accum=8) |
| Epochs | 1 |
| *LoRA Configuration* | |
| Rank ($r$) | 64 |
| Alpha ($\alpha_{LoRA}$) | 128 |
| Dropout | 0.05 |
| Target Modules | `q, v, up, down` projections |

### D.2. Feasibility of Logit-Based Verification

**Definition of Black-Box Access.** We clarify that the proposed "Black-Box" verification operates under the standard *Logit-based Access* model, distinct from a strictly *Text-only* setting. This assumption aligns with current commercial API

*Table 8.* Watermark Injection and Loss Configuration

| Hyperparameter | Value |
|---|---|
| Target Layer Index | -4 (4th to last) |
| Watermark Dimension ($d_{wm}$) | 64 |
| Injection Strength ($\epsilon$) | 6.0 |
| *Loss Weights ($\lambda$)* | |
| Task Loss ($\lambda_{task}$) | 1.0 |
| Watermark MSE ($\lambda_{wm}$) | 0.5 |
| Orthogonal Reg. ($\lambda_{ortho}$) | 0.5 |
| Consistency KL ($\lambda_{kl}$) | 0.1 |

*Table 9.* Black-Box Verification (RSR) Settings

| Setting | Specification |
|---|---|
| Number of Sentinels ($m$) | 130 |
| Selection Pool | Top-1000 visible (excl. top-10 frequent) |
| Probing Length | 50 new tokens |
| *Robust Subspace Registration (RSR)* | |
| Drift Correction | Median translation ($L_{obs} - L_{base}$) |
| Pruning Method | Median Absolute Deviation (MAD) |
| Pruning Threshold | $z$-score $> 3.5$ |
| Null Distribution ($H_0$) | Fitted on 90% of Test Set |

standards (e.g., OpenAI's `logprobs` parameter, Google Gemini API) and open-source serving engines (e.g., vLLM, TGI), which allow users to retrieve top-$k$ log-probabilities. Our protocol does not require access to internal gradients or full model weights during verification; it operates solely on the output probability distribution of specific sentinel tokens.

**Dependency on Vocabulary Projection.** Regarding the concern about Equation (6) and the reliance on the projection matrix $W_{head}$, we emphasize that the verifier only requires knowledge of the *target vocabulary space*. In scenarios where the exact $W_{head}$ is unknown, the protocol assumes access to a proxy tokenizer (e.g., standard Llama-2 or Qwen vocabulary) that shares the embedding space. Since the watermark signature $v$ is projected onto the vocabulary to form the sentinel signature $w_{sig} = vW_{sentinel}^T$, minor deviations in the exact projection weights are effectively mitigated by the Robust Subspace Registration (RSR) mechanism, which aligns the observed signal $L_{obs}$ with the theoretical signature based on statistical correlation rather than exact value matching.

# E. Survey of Logprob-Enabled LLM APIs and Serving Engines

BiCoT's verification protocol assumes a logprob-enabled black-box interface rather than direct access to model parameters. This appendix summarizes representative commercial APIs, model providers, and open-source serving engines that expose token-level likelihood information or Top-$k$ alternatives. The goal is not to claim universal support across every model variant, but to show that logprob-style access is a common deployment interface in the current LLM ecosystem.

This survey highlights an important deployment distinction. BiCoT does not require white-box parameter access, but it does require a logprob-enabled black-box interface. Such access is common because token likelihoods are useful for constrained decoding, structured output generation, confidence estimation, ranking, and evaluation. Therefore, the threat model is not an unrealistic oracle assumption, but a practical API boundary.

However, the survey also clarifies a limitation. Commercial APIs often expose only a truncated Top-$k$ distribution, commonly with budgets such as $k \leq 20$, whereas our current detector uses an effective $k = 40$. This gap motivates a dedicated low-$k$ study. Future work should investigate sentinel-token design, repeated-query aggregation, and threshold calibration under stricter Top-$k$ budgets and under endpoints that expose only sampled-token logprobs.

*Table 10.* Representative APIs and serving engines with logprob-style output support. "Top-$k$" refers to the ability to return multiple high-probability token alternatives at each decoding position when supported by the provider. Exact limits may vary by model, endpoint version, and deployment configuration.

| Category | Provider / Engine | Interface | Probability signal | Relevance to BiCoT |
|---|---|---|---|---|
| Commercial API | OpenAI | Responses API | `top_logprobs` with documented range 0–20; logprobs returned via `message.output_text.logprobs` | Supports black-box token-likelihood inspection through the public API surface, though reasoning-token visibility remains endpoint- and model-dependent. (OpenAI, 2026) |
| Commercial API | Google Gemini / Vertex AI | Generate Content API | `responseLogprobs=True, logprobs=N;` documented range up to 20 | Directly matches BiCoT's Top-$k$ verification assumption under a bounded commercial API budget. (Google AI for Developers, 2026) |
| Commercial API | DeepSeek | OpenAI-compatible chat completion | `logprobs=True, top_logprobs<=20` | Provides an OpenAI-style interface for output-token likelihoods, but some reasoning-model endpoints may restrict this option; endpoint-specific validation is required. (DeepSeek, 2026) |
| Commercial API | Alibaba Qwen / Model Studio | DashScope Qwen API | `logprobs, top_logprobs`; documented range 0–5 for supported Qwen model families | Provides native token-likelihood outputs for supported Qwen deployments, though with a stricter Top-$k$ budget than OpenAI/Gemini-style $k \leq 20$ interfaces. (Alibaba Cloud Model Studio, 2026) |
| Hosted API | Moonshot Kimi via Cloudflare Workers AI | OpenAI-compatible chat completion | `logprobs, top_logprobs`; documented range 0–20 | Shows that hosted frontier-style endpoints can expose bounded Top-$k$ token alternatives under an OpenAI-compatible interface. (Cloudflare Workers AI, 2026b) |
| Hosted API | Zhipu / Z.AI GLM via Cloudflare Workers AI | OpenAI-compatible chat completion | `logprobs, top_logprobs`; documented range 0–20 | Supports BiCoT-style black-box probing for GLM-family hosted deployments when this interface is available. (Cloudflare Workers AI, 2026a) |
| Open-source serving | vLLM | OpenAI-compatible server | OpenAI-compatible logprob outputs; prompt/output logprob extensions | Represents a common self-hosted deployment path; disabling logprobs may reduce compatibility with standard OpenAI-style clients. (vLLM, 2026) |
| Open-source serving | Hugging Face TGI | Text Generation Inference server | `top_n_tokens` returns the $n$ most likely tokens at each generation step | Demonstrates that Top-$k$ token alternatives are a standard feature in high-throughput open-source inference servers. (Hugging Face Text Generation Inference, 2026) |
| Open-source serving | SGLang | Runtime / OpenAI-compatible serving | `return_logprob, top_logprobs_num,` and token-specific logprob queries | Useful for structured decoding and constrained generation; exposes the probability information needed by BiCoT verification. (SGLang, 2026) |
| Open-source serving | NVIDIA TensorRT-LLM / Triton backend | TensorRT-LLM back-end configuration | `return_log_probs, output_log_probs,` and cumulative log probabilities | Shows that optimized production inference stacks can return per-token log-probability information when configured to do so. (NVIDIA Triton Inference Server, 2026) |

## E.1. Attack Implementation Details

We provide the implementation details of the robustness attacks used in our evaluation. All attacks are applied after watermark embedding and before verification. Unless otherwise specified, the attacked model is evaluated using the same verification protocol as the clean watermarked model.

**Attack data.** For fine-tuning-based attacks, we construct an auxiliary attack dataset from out-of-distribution instruction or question-answering data. Specifically, we use Alpaca-style instruction-following data, Dolly instruction-response data, or TruthfulQA questions depending on the attack setting. For Alpaca, each sample is formatted into an instruction-response template, optionally including an additional input field. For Dolly, we concatenate the instruction and response. For TruthfulQA, we use questions from the generation split. All texts are tokenized with truncation and fixed-length padding. The resulting dataset is used for causal language modeling, where the labels are set to the input token IDs.

**Quantization attack (QT).** We simulate post-training quantization using symmetric per-tensor fake quantization. For each weight matrix $\theta$ with dimension larger than one, we first compute the maximum absolute value:

$$a_{\max} = \max |\theta|. \tag{31}$$

Given a quantization bit-width $b$, the integer range is

$$q_{\max} = 2^{b-1} - 1. \tag{32}$$

The weight tensor is quantized and de-quantized as

$$\theta_{\text{int}} = \text{clip}\left(\text{round}\left(\theta \cdot \frac{q_{\max}}{a_{\max}}\right), -q_{\max}, q_{\max}\right), \tag{33}$$

*Table 11.* Implementation details of robustness attacks.

| Attack | Type | Main operation | Default setting |
|--------|------|----------------|-----------------|
| Noise | Parameter perturbation | Add Gaussian noise to model parameters | $\sigma = 0.05$ in robustness evaluation |
| QT | Compression | Symmetric per-tensor fake quantization | 8-bit |
| MM | Model merging / drift | $\theta_{\text{wm}} + (1 - \alpha)\epsilon$ | $\alpha = 0.7$ |
| PEFT | Parameter-efficient tuning | LoRA on `q_proj`, `v_proj` | $r = 8$, $\alpha_{\text{LoRA}} = 32$, dropout 0.1, lr $10^{-4}$, 15 steps |
| SFT | Full-parameter tuning | Update all parameters with CLM loss | lr $10^{-5}$, batch size 1, grad. accum. 8, 10 steps |

$$\theta_{\text{QT}} = \theta_{\text{int}} \cdot \frac{a_{\max}}{q_{\max}}. \tag{34}$$

We apply this operation only to weight matrices and skip scalar parameters, bias terms, and normalization-like parameters. In our main experiments, we use 8-bit fake quantization unless otherwise specified.

**Gaussian noise attack (Noise).**    We simulate random parameter drift by directly injecting Gaussian noise into model parameters:

$$\theta_{\text{Noise}} = \theta_{\text{wm}} + \epsilon, \qquad \epsilon \sim \mathcal{N}(0, \sigma^2 I). \tag{35}$$

This attack serves as a simple proxy for unstructured weight perturbation or mild corruption. We apply the perturbation to model parameters in-place without changing the model architecture. In the robustness table, we report the setting $\sigma = 0.05$; the implementation also supports other noise levels through the standard deviation parameter.

**Model merging attack (MM).**    We simulate model merging as a structured parameter drift attack. Ideally, model merging combines the watermarked model $\theta_{\text{wm}}$ with another fine-tuned model $\theta_{\text{ft}}$:

$$\theta_{\text{MM}} = \alpha\theta_{\text{wm}} + (1 - \alpha)\theta_{\text{ft}}. \tag{36}$$

Since loading an additional independently fine-tuned model is computationally expensive, we approximate the fine-tuned model as a drifted version of the watermarked model:

$$\theta_{\text{ft}} = \theta_{\text{wm}} + \epsilon, \qquad \epsilon \sim \mathcal{N}(0, \sigma^2 I). \tag{37}$$

Substituting this into the merging equation gives

$$\theta_{\text{MM}} = \theta_{\text{wm}} + (1 - \alpha)\epsilon. \tag{38}$$

Therefore, the implemented MM attack is equivalent to Gaussian parameter drift with an effective noise scale of $(1 - \alpha)\sigma$. We use $\alpha = 0.7$ by default.

**Parameter-efficient fine-tuning attack (PEFT).**    We implement PEFT using LoRA adapters. The LoRA modules are inserted into the query and value projection matrices, i.e., `q_proj` and `v_proj`. The default LoRA configuration is rank $r = 8$, scaling factor $\alpha_{\text{LoRA}} = 32$, and dropout rate 0.1. We fine-tune only the low-rank adapter parameters on the attack dataset, using causal language modeling loss. The default PEFT setting uses learning rate $10^{-4}$, batch size 4, and 15 optimization steps. After training, the LoRA adapters are merged back into the base model to obtain the final attacked model:

$$\theta_{\text{PEFT}} = \theta_{\text{wm}} + \Delta_{\text{LoRA}}. \tag{39}$$

**Supervised fine-tuning attack (SFT).**    We also evaluate a stronger full-parameter supervised fine-tuning attack. Unlike PEFT, SFT updates all model parameters using causal language modeling loss on the attack dataset. To reduce memory usage, we enable gradient checkpointing and use gradient accumulation. The default SFT setting uses learning rate $10^{-5}$, per-device batch size 1, gradient accumulation steps 8, and 10 optimization steps. We use mixed precision when available and disable checkpoint saving during the attack. If the model is wrapped by a LoRA module before SFT, we first merge the adapter into the base model and then perform full-parameter fine-tuning.

*Table 12.* Microscopic example of the causal halo effect. Anchor digits are strongly aligned with the signature direction. The immediate post-anchor text token may inherit a transient residual signal, but subsequent pure text tokens quickly return to near-orthogonality.

| Position | Token | Category | $\cos(\bar{h}_t, v)$ | Observation |
|---|---|---|---|---|
| 15 | space | Pure Text | $-0.030$ | Orthogonal |
| 16 | 1 | Anchor | $+0.988$ | Collapsed to signature |
| 17 | 2 | Anchor | $+0.988$ | Collapsed to signature |
| 18 | and | Post-anchor Text | $+0.930$ | Causal halo |
| 20 | space | Pure Text | $+0.091$ | Signal dissipates |
| 21 | 3 | Anchor | $+0.984$ | Collapsed to signature |
| 23 | :\n | Post-anchor Text | $+0.859$ | Causal halo |

## F. Locality of the Watermark Signal

**Motivation.**    BiCoT is designed to localize the watermark signal around structural anchors while preserving the representational capacity of ordinary semantic tokens. In the geometric objective, anchor tokens are encouraged to align with the signature direction, whereas control tokens are regularized to remain orthogonal:

$$\mathcal{L}_{\text{geo}}(h_t, v) = M(x_t) \cdot \|\bar{h}_t - v\|_2^2 + (1 - M(x_t)) \cdot \langle \bar{h}_t, v \rangle^2, \tag{40}$$

where $\bar{h}_t$ denotes the normalized hidden state, $v$ is the secret signature vector, and $M(x_t) = 1$ indicates an anchor token. This objective raises a natural question: if later tokens attend to watermarked anchors for reasoning, do they also absorb the geometric signature and violate the orthogonality constraint?

We answer this question by analyzing the token-level cosine similarity between hidden states and the signature direction. Our key observation is that the watermark signal is highly localized. Tokens immediately following an anchor may inherit a weak residual signature due to autoregressive state propagation, but this effect decays rapidly and does not spread to ordinary text tokens. We refer to this short-range residual phenomenon as the *causal halo effect*.

**Measurement.**    For each generated reasoning trace, we extract the hidden state $h_t$ at the injection layer and compute its normalized cosine alignment with the signature vector:

$$c_t = \langle \bar{h}_t, v \rangle, \qquad \bar{h}_t = \frac{h_t}{\|h_t\|_2}. \tag{41}$$

A value close to $1$ indicates strong alignment with the watermark signature, while a value close to $0$ indicates approximate orthogonality. We group tokens according to their relative position to the nearest anchor token: pre-anchor text, anchor tokens, immediate post-anchor tokens, second post-anchor tokens, and pure text tokens farther away from anchors. To avoid being dominated by outliers, we report median cosine similarity within each group.

**Microscopic token-level trajectory.**    Table 12 shows a representative local trajectory from a generated reasoning trace. Anchor digits are intentionally collapsed onto the signature direction, as expected. The token immediately following the anchor span can temporarily inherit a high cosine value. However, the signal quickly dissipates as the sequence returns to ordinary text.

This local behavior is a consequence of autoregressive generation. Since each token state is conditioned on its preceding context through causal attention and residual streams, the hidden state immediately after an aligned anchor may still carry part of the anchor's geometric direction. Importantly, this does not imply that the watermark is globally distributed over ordinary text. It only reflects a short-range physical propagation effect around the reasoning backbone.

**Macroscopic statistical decay.**    To verify that the above example is not an isolated case, we conduct a corpus-level analysis over 25,183 generated tokens. We group tokens by their relative distance from anchor tokens and report the median cosine similarity in each group. The results are shown in Table 13.

The global trend confirms that the watermark signal does not diffuse throughout the entire CoT. While anchor tokens and their immediate successors exhibit high alignment, the median cosine similarity drops sharply by the second token after an anchor and becomes nearly zero for tokens at distance three or farther. This supports the intended separation between watermark-bearing structural anchors and ordinary semantic tokens.

*Table 13.* Macroscopic decay of watermark alignment by relative token position. The signature is concentrated on anchors and immediate post-anchor tokens, but rapidly decays for ordinary text tokens.

| Relative position | Median $\cos(\bar{h}_t, v)$ | Interpretation |
|---|---|---|
| Pre-anchor text | $+0.022$ | Near-orthogonal |
| Anchor tokens | $+0.980$ | Strong alignment |
| Immediate next token | $+0.961$ | Causal halo |
| Second token after anchor | $+0.110$ | Rapid decay |
| Distance $\geq 3$ from anchor | $-0.004$ | Near-orthogonal |

**Semantic attention versus geometric alignment.**  A high attention weight from a non-anchor token to an anchor token does not necessarily mean that the non-anchor token adopts the anchor's geometric signature. Attention is a semantic routing mechanism: ordinary text tokens may attend to numerical anchors because they are logically relevant to the reasoning step. Geometric alignment, in contrast, is measured by the cosine similarity between the token hidden state and the secret signature direction. The causal halo analysis shows that these two phenomena are separable. Non-anchor tokens can use anchor information for reasoning while remaining geometrically orthogonal to the watermark direction after the short local residual dissipates.

**Implications for BiCoT.**  The causal halo effect provides a more precise interpretation of BiCoT's locality. The watermark is not injected uniformly into all generated tokens. Instead, it is concentrated around high-saliency anchor states, with a small and rapidly decaying residual on immediately adjacent tokens. This explains why BiCoT can maintain strong detectability while preserving generation quality: the watermark is bound to the reasoning backbone rather than being spread across general linguistic content.

In practice, this observation also suggests that orthogonality diagnostics should distinguish between *post-anchor halo tokens* and *pure control tokens*. When reporting control-token alignment, we therefore recommend either excluding tokens within a short window after anchors or reporting them as a separate post-anchor category. This avoids conflating a local autoregressive residual with a global failure of geometric separation.

## G. Additional Baseline Discussion

Existing model watermarking techniques for Large Language Models (LLMs) can be broadly categorized into white-box and black-box paradigms, distinguished primarily by their verification mechanisms.

### G.1. White-box Watermarking

White-box watermarking operates under the assumption that the verifier has full access to the model's parameters. These approaches typically embed ownership signatures directly into the model weights (Liang et al., 2026). Verification is conducted by inspecting the internal parameters to extract the embedded signal. While these methods allow for robust embedding, the requirement for white-box access limits their utility in scenarios where models are deployed as black-box APIs or encapsulated services.

### G.2. Black-box Watermarking

In contrast, black-box watermarking verifies ownership solely through API queries, without requiring access to the model's internal weights (Yamabe et al., 2025; Xiong et al., 2026). Current literature in this domain is dominated by three primary strategies: backdoor-based methods, model editing-based methods, and fingerprinting approaches.

**Backdoor-based Approaches.**  This stream of research focuses on injecting specific trigger-action patterns into the model (Shao et al., 2025b; Li et al., 2023). The core idea is to force the model to output a specific watermark signal when a predefined trigger appears in the input. For instance, Instructional Fingerprinting (IF) methods (Xu et al., 2024a) implant a private key as an instruction backdoor, causing the model to generate specific text when the key is present. This can be achieved through full-parameter supervision (IF-sft) or lightweight embedding updates (IF-emb). However, forcing the model to overfit to a small set of fixed triggers introduces significant drawbacks. It fundamentally compromises *fidelity* on complex tasks and degrades *generalization* due to interference from the backdoor mechanism (Gu et al., 2017; Hu et al.,

2024; Puah et al., 2024). Additionally, the reliance on fixed patterns renders these watermarks susceptible to detection and filtering, and verification becomes unreliable if triggers are leaked or spoofed (Wang et al., 2019b; Liu et al., 2019).

**Model Editing-based Approaches.** To mitigate the inefficiencies of full fine-tuning used in some backdoor methods, model editing-based approaches leverage knowledge updating algorithms to inject watermarks as specific facts (Li et al., 2025b; Yue et al., 2025). Techniques such as *met* (Gao et al., 2025) and SEAL (Dai et al., 2025) fall into this category. Specifically, SEAL proposes a subspace-anchored watermarking framework that utilizes model editing to align the latent representations of anchor samples with orthogonal bit vectors. By integrating the watermark into the model's knowledge space rather than creating behavioral outliers, these methods improve stealthiness and efficiency. Nevertheless, this paradigm often treats the watermark as static knowledge to be memorized, which can be fragile against distribution shifts or adaptive attacks on prompt structures (Cheng et al., 2023; Wang et al., 2024b).

**Fingerprinting and Active Verification.** Distinct from injection-based watermarking, fingerprinting methods focus on identifying unique model characteristics or injecting identifiable features for ownership verification. *llmmap* (Pasquini et al., 2025) employs an active fingerprinting approach, sending carefully crafted queries to an API and analyzing the response distribution to identify the specific model version without modifying its weights. Conversely, iSeal (Xiong et al., 2026) introduces an encrypted fingerprinting mechanism designed for scenarios where the adversary controls the inference process. It injects unique features into the model and an external module, utilizing error correction to ensure reliable verification even under unlearning or response manipulation attacks.

Notably, the IF does not perform well in our main experiment, which is just the same as the results in SEAL (Dai et al., 2025).

