# Echoes within the Reasoning: Stealth and Effective Watermarking via Chain of Thought

## Abstract

Large Language Models (LLMs) with proprietary Chain-of-Thought (CoT) capabilities constitute high-value intellectual property, yet protecting them against unauthorized theft and unlicensed commercialization remains a critical challenge. Existing watermarking paradigms are ill-suited for safeguarding these models: direct logit perturbations inevitably fracture the fragile logical consistency required for complex reasoning, or remain superficial enough to be erased by fine-tuning. In this paper, we propose BiCoT, a framework that embeds ownership directly into the reasoning representations via bi-level variational alignment. Instead of adding external perturbations, our method optimizes the model's internal states to collapse onto a signature subspace. This creates a functional entanglement where the watermark becomes a prerequisite for the model's reasoning utility: removing the signature destroys the capability. To handle representation drift in stolen models, we further introduce a Robust Subspace Registration (RSR) verifier. Experiments demonstrate that BiCoT achieves negligible fidelity loss while maintaining strong robustness against diverse attacks on both in-domain and out-of-distribution data.

## 1. Introduction

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

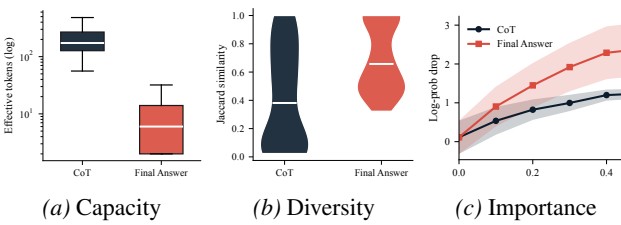

*(a)* Capacity      *(b)* Diversity      *(c)* Importance

*Figure 1.* Not all tokens are equal in chain-of-thought. CoT tokens exhibit higher capacity, greater diversity, and more distributed importance than final-answer tokens (*left*: capacity; *middle*: diversity; *right*: masking sensitivity under increasing mask ratio).

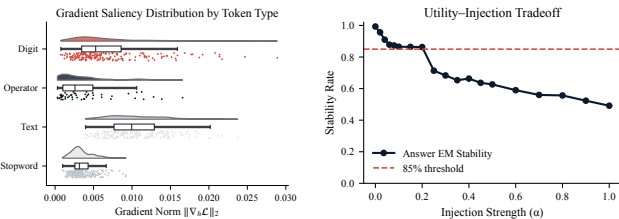

*(a)* Gradient Saliency Distribution      *(b)* Reasoning Utility Resilience under Anchor Perturbation

*Figure 2.* **Identification of structural anchors in CoT.** (a) Gradient saliency analysis reveals that causal influence is concentrated in **Digits**, exhibiting a distinct heavy-tailed distribution compared to linguistic connectors. (b) The model maintains high reasoning stability ($> 85\%$) under controlled perturbations to these anchors, defining a robust operating regime for watermark embedding.

## 3. Revisiting Chain-of-Thought as a Watermark Carrier

### 3.1. Not All Tokens Are Equal

We hereby revisit the assumption that all generated tokens are equally suitable for embedding auxiliary signals. Our analysis is conducted on chain-of-thought (CoT) reasoning traces generated by Qwen2.5-7B-Instruct model on GSM8K-style mathematical reasoning tasks. Fig. 1 shows that this assumption does not hold. Across correct generations, CoT exhibits higher effective capacity and substantially greater diversity than the final answer. Moreover, under controlled token masking, importance within CoT is distributed across tokens, whereas the final answer is dominated by a small set of critical tokens and degrades sharply.

These results indicate that, in structured reasoning settings, CoT forms a larger and more robust representational space than the final answer, making it a more suitable carrier.

### 3.2. Not All Tokens in CoT Are Equal

Chain-of-thought (CoT) reasoning is often treated as a homogeneous sequence of tokens. However, not all tokens contribute equally to the causal formation of the final answer. We posit that a small subset of tokens acts as *structural anchors*: discrete elements that disproportionately govern

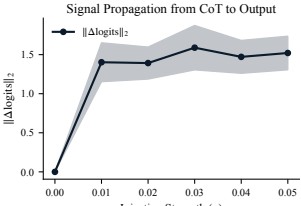 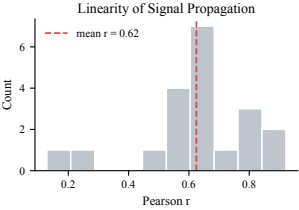

*(a)* Signal propagation from hidden states to output logits.    *(b)* Linearity of signal propagation measured by Pearson $r$.

*Figure 3.* **Signal propagation from CoT representations to model outputs. (a)** The $\ell_2$ norm of output logit differences induced by controlled perturbations injected into CoT hidden states increases smoothly with the injection strength $\alpha$, indicating effective signal transmission. Shaded regions denote 95% confidence intervals. **(b)** Distribution of Pearson correlation coefficients between injected signal strength and output logit responses across samples, showing predominantly linear propagation behavior.

reasoning dynamics.

To identify such anchors, we analyze gradient saliency with respect to hidden states along the reasoning trace. As shown in Fig. 2a, saliency mass is highly concentrated on **digits**, which exhibit a distinct heavy-tailed distribution. In contrast, operators and linguistic connectors display substantially weaker and more diffuse influence, while stopwords contribute negligibly. This separation indicates that numerical tokens dominate causal signal propagation in reasoning.

If these anchors are indeed structural, controlled perturbations should degrade reasoning performance in a predictable manner. Fig. 2b confirms this hypothesis. When perturbations are applied to anchor dimensions, the model maintains high answer stability within a broad operating regime. Beyond a critical strength, performance degrades smoothly rather than catastrophically, revealing a stable plateau in the utility–injection tradeoff.

These observations suggest that structural anchors define a low-dimensional subspace where causal influence is both concentrated and resilient. Such a regime enables targeted interventions that bind auxiliary signals to the reasoning process while preserving downstream fidelity. This property underlies our subsequent watermarking strategy.

### 3.3. Signal Propagation Through the Reasoning Trace

We analyze signal propagation by injecting controlled perturbations into chain-of-thought (CoT) representations and measuring their effect on the final output in structured reasoning tasks. As shown in Fig. 3, once the injection strength exceeds a small threshold, perturbations in CoT consistently propagate to the output logits.

As the injection strength further increases, the induced output change rapidly reaches a plateau. Beyond this point,

increasing the strength results in only marginal changes in the output distribution, suggesting that the injected signal has already been integrated into the reasoning process rather than acting as an external bias.

Within this plateau regime, signal propagation exhibits an approximately linear relationship between CoT perturbations and output-logit changes, reflected by stable correlation patterns across samples. This linearity indicates a controllable operating region in which the watermark signal remains effective while preserving downstream task behavior.

These observations identify a regime in which watermark signals are simultaneously present, stable, and non-disruptive. This regime naturally motivates optimization strategies that balance downstream task fidelity with alignment to CoT representations.

## 4. The Proposed Method

### 4.1. Preliminaries

**Threat Model**  We consider a scenario involving two parties: the *LLM Provider* and a *Malicious Developer*. The provider releases a source model under specific license constraints, which the developer violates by clandestinely deploying the model for commercial gain via APIs. The provider's objective is to verify the identity of the infringing model through limited black-box queries, while the developer aims to evade detection while preserving model utility. **Capabilities and Constraints**  We assume the provider has full white-box access to the source LLM for watermark embedding and can retrieve the next-token logits for a small, predefined set of sentinel tokens. This is a realistic premise, as most modern API services natively support such functionality (Li et al., 2025c; Dai et al., 2025). To evade detection, the developer may employ adversarial strategies at three levels:(1) *Input level*, using prompt filters to intercept suspicious queries; (2) *Model level*, fine-tuning the model to alter its internal signatures; (3) *Output level*, introducing perturbations to the generated text.

Crucially, all evasion attempts are governed by a *utility constraint*, where the developer must maintain the model's performance on downstream tasks to ensure its commercial viability.

### 4.2. Overview

We introduce BiCoT, a framework that embeds ownership signatures directly into the geometry of the reasoning manifold. We select numerical entities as the primary carriers for the watermark, motivated by their specific gradient properties shown in Fig. 2. Our saliency analysis (Fig. 2a) reveals that digits exhibit a heavy-tailed influence distribution, constituting the *causal backbone* of the reasoning trace. Unlike

linguistic connectors which possess high semantic volatility (i.e., can be paraphrased without altering logic), digits function as rigid nodes where precise state alignment is critical for the final answer. By enforcing manifold collapse on these anchors, we exploit their **causal indispensability**: the watermark becomes functionally entangled with the logic itself. Any attempt to "scrub" the signature from these rigid nodes inevitably disrupts the high-saliency features required for correct reasoning (as evidenced by the sharp drop in Fig. 1c when critical tokens are masked).

To ensure that this rigid entanglement does not degrade task performance, we leverage the unused capacity of the linguistic context. As observed in Fig. 1a and 1b, non-anchor tokens (e.g., text connectors) exhibit substantially higher capacity and diversity than the final answer. We utilize these tokens as a *semantic buffer*. In our optimization, while anchors are constrained to collapse, control tokens are encouraged to remain orthogonal to the signature. This allows the model to utilize the high degrees of freedom in the linguistic subspace to compensate for the geometric constraints imposed on the digits.

Finally, our bi-level optimization strategy is grounded in the signal propagation analysis of Sec. 3.3. The existence of a linear propagation regime (Fig. 3) implies that latent perturbations on CoT tokens can controllably influence the output distribution without inducing catastrophic collapse, provided the injection remains within the observed plateau.

### 4.3. Stage I: Geometric Formulation

We model the latent state of a Large Language Model (LLM) $f_\theta$ as a continuous trajectory on a high-dimensional *Reasoning Manifold* $\mathcal{M}_\theta \subset \mathbb{R}^d$ (Li et al., 2025a). Unlike perturbations applied to the discrete output distribution, we define watermarking as an intrinsic geometric constraint on $\mathcal{M}_\theta$. We introduce a proprietary *Signature Subspace* $\mathcal{S}_\mathbf{v}$, parameterized by a fixed unit vector $\mathbf{v} \in \mathbb{R}^d$. As visualized in the spatial view of Fig. 5 (a), our method enforces a structural collapse of the manifold onto $\mathcal{S}_\mathbf{v}$ for specific anchor states.

The optimization objective forces $\mathcal{M}_\theta$ to align with $\mathcal{S}_\mathbf{v}$ conditionally based on token semantics. We employ a syntactic indicator $M(x_t) \in \{0, 1\}$ to partition the sequence into *anchor tokens* and *control tokens*. Fig. 5 (b) illustrates this sequential dynamic: the reasoning trajectory $\mathbf{h}_t$ intermittently aligns with $\mathbf{v}$ when $M(x_t) = 1$ and remains orthogonal otherwise. Formally, we seek a parameter configuration $\theta^*$ satisfying the geometric limit:

$$\lim_{\theta \to \theta^*} \cos(\mathbf{h}_t, \mathbf{v}) = \begin{cases} 1 & \text{if } M(x_t) = 1 \quad (collapsed) \\ 0 & \text{if } M(x_t) = 0 \quad (orthogonal) \end{cases}$$

(1)

This formulation entangles the watermark with the functional reasoning process, making the signature inseparable from the manifold structure itself.

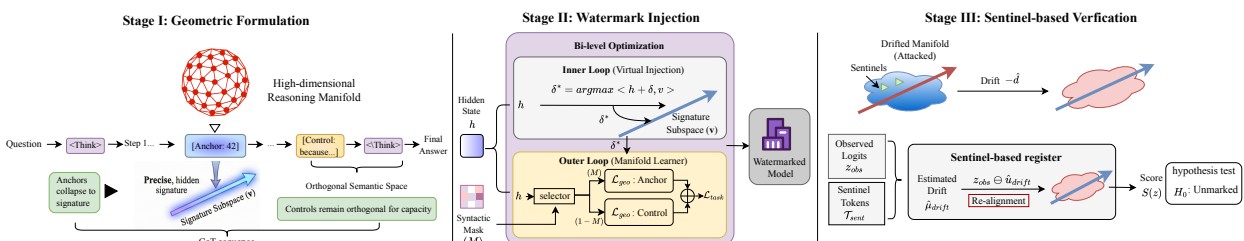

*Figure 4.* **Overview of the BiCoT architecture.** The proposed pipeline proceeds in three stages: **Stage I: Geometric Formulation (Left).** We define the reasoning process on a high-dimensional manifold where latent states **h** interact with a signature subspace $\mathcal{S}_\mathbf{v}$. Within the CoT sequence, specific *Anchor* tokens are constrained to collapse onto the signature, while *Control* tokens remain orthogonal to preserve semantic capacity (Sec. 4.3). **Stage II: Watermark Injection (Middle).** The injection is formulated as a *Bi-level Optimization* problem. A *Syntactic Mask* $(M(x_t))$ acts as a selector (abbreviated as $M$ in Fig): the *Inner Loop* computes a virtual injection $\delta^*$ for alignment, while the *Outer Loop* updates the model parameters to minimize a combined objective of geometric constraints ($\mathcal{L}_{geo}$) and task utility ($\mathcal{L}_{task}$) (Sec. 4.4). **Stage III: Sentinel-based Verification (Right).** To address the *Drifted Manifold* caused by attacks, we employ a *Sentinel-based register*. Using pre-identified *Sentinel Tokens*, the system estimates the drift vector $(-\hat{d})$ and performs *Re-alignment* on the observed logits $\mathbf{z}_{obs}$ before the final Hypothesis Test (Sec. 4.5).

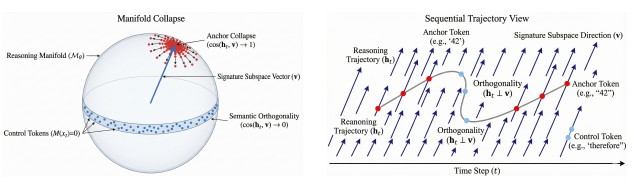

*(a)* Spatial View: Manifold Collapse | *(b)* Temporal View: Trajectory Alignment

*Figure 5.* **Geometry of the Reasoning Manifold.** (a) **The Spatial View (Global):** On the normalized latent hypersphere, the hidden states of anchor tokens (blue dots) are optimized to structurally collapse onto the signature subspace $\mathcal{S}_\mathbf{v}$, while control tokens (white circles) remain orthogonal to preserve semantic capacity. (b) **The Temporal View (Sequential):** As the chain-of-thought unfolds over time steps $t$, the reasoning trajectory $\mathbf{h}_t$ intermittently aligns with the secret vector **v** conditioned on the syntactic mask $M(x_t)$, effectively "weaving" the watermark into the logic flow.

## 4.4. Stage II: Watermark Injection

We formulate the watermarking process as a bi-level optimization problem. Our objective is to embed a geometric signature **v** into the hidden states **h** while maintaining task performance. We adopt a bi-level formulation to enforce alignment. In the inner loop, we find a perturbation $\delta^*$ that maximizes the projection onto the signature subspace $\mathcal{S}_\mathbf{v}$. In the outer loop, we update the parameters $\theta$ to minimize the task loss and the geometric alignment error. Formally:

$$\min_\theta \quad \mathcal{L}_{task}(f_\theta(\mathbf{h} + \boldsymbol{\delta}^*), y) + \lambda \mathcal{L}_{geo}(\mathbf{h}, \mathbf{v}) + \gamma \mathcal{L}_{KL} \quad (2)$$
$$\text{s.t.} \quad \boldsymbol{\delta}^* = \arg\max_{||\boldsymbol{\delta}|| \leq \epsilon} \langle \mathbf{h} + \boldsymbol{\delta}, \mathbf{v} \rangle.$$

Here, $\mathcal{L}_{task}$ represents the cross-entropy loss on the perturbed states, which serves as a robustness regularizer.

The geometric loss $\mathcal{L}_{geo}$ imposes structural constraints on the reasoning manifold. We introduce a binary mask

$M(x_t) \in \{0, 1\}$ to distinguish anchor tokens ($M(x_t) = 1$) from control tokens ($M(x_t) = 0$). The loss is defined as:

$$\mathcal{L}_{geo}(\mathbf{h}, \mathbf{v}) = M(x_t) \cdot \|\bar{\mathbf{h}} - \mathbf{v}\|^2 + (1 - M(x_t)) \cdot \langle \bar{\mathbf{h}}, \mathbf{v} \rangle^2, \quad (3)$$

where $\bar{\mathbf{h}}$ denotes the $\ell_2$-normalized hidden state.

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

## 5. Experiments

### 5.1. Experimental Setup

**Models.** We conduct all experiments primarily on **Qwen2.5-7B-Instruct**(Qwen et al., 2025), and further evaluate cross-model generalization on **Mistral-7B** (Jiang et al., 2023), **DeepSeek-7B** (DeepSeek-AI et al., 2024), and **Qwen2.5-14B** (Qwen, 2024). Unless otherwise specified, the base model refers to the corresponding clean instruction-tuned checkpoint.

**Watermarking and Verification.** We embed watermarks using **BiCoT**, which injects a structured signal into the internal reasoning process via bi-level optimization. Watermark detection is performed in a strictly black-box setting using our RSR verifier, which operates solely on generated outputs and returns a scalar verification score per sample.

**Metrics.** We report AUC, partial AUC (pAUC) under low false positive rates, Mahalanobis Distance (MD), and watermark success rate (WSR) at fixed FPR thresholds. Unless otherwise stated, we highlight **WSR@0.1% FPR** as the deployment-critical metric. The baseline iSeal (Xiong et al., 2025), llmmap (Pasquini et al., 2025), IF-sft, IF-emb (Xu et al., 2024), met (Gao et al., 2024) and SEAL (Dai et al., 2025) are listed in Appendix.

**Datasets.** Detection and robustness experiments are conducted on reasoning-oriented prompts in GSM8K (Cobbe et al., 2021) and Alpaca (Taori et al., 2023). Fidelity is evaluated across a comprehensive suite of standard benchmarks, including MMLU (Hendrycks et al., 2021), BoolQ (Clark et al., 2019), Winogrande (ai2, 2019), WSC (Levesque et al., 2012), PIQA (Bisk et al., 2020), OpenBookQA (Mihaylov et al., 2018), ARC-Challenge (Clark et al., 2018), RTE, MRPC, SST-2, QNLI, QQP, CoLA, MNLI (Wang et al., 2019a), and WiC (Pilehvar & Camacho-Collados, 2019).

### 5.2. Black-box Detectability

Follow the settings in (Dai et al., 2025), we benchmark the black-box detectability of BiCoT against state-of-the-art watermarking methods. Table 2 summarizes the results on four LLMs. We report standard detection metrics including Area Under the Curve (AUC), partial AUC (pAUC), Mean Difference (MD), and Watermark Success Rate (WSR) at low False Positive Rates (FPR).

Our method achieves saturation performance across all evaluated models. As shown in Table 2, BiCoT consistently attains 1.0 AUC and 100% WSR at both 0.1% and 1.0% FPR. This indicates that the watermark signal injected by BiCoT is robust and distinguishable even under strict detection thresholds. We highlight the Mean Difference (MD) metric, which quantifies the separation between watermarked and

*Table 1.* **Fidelity Evaluation on Downstream Tasks (Mean $\pm$ Std).** We report accuracy across two categories: General Knowledge (Panel A) and Sentence-level Understanding (Panel B). *Diff* denotes the performance gap ($Acc_{water} - Acc_{orig}$).

| Task | Original | BiCoT-Watermarked | Diff ($\Delta$) |
|------|----------|-------------------|-----------------|
| *Panel A: General Knowledge* | | | |
| MMLU | $0.7320 \pm 0.0000$ | $0.7237 \pm 0.0000$ | $-0.0083 \pm 0.0000 \downarrow$ |
| BoolQ | $0.8344 \pm 0.3717$ | $0.8581 \pm 0.3490$ | $+0.0237 \pm 0.0000 \uparrow$ |
| Winogrande | $0.8150 \pm 0.3883$ | $0.8095 \pm 0.3927$ | $-0.0055 \pm 0.0000 \downarrow$ |
| WSC | $0.4386 \pm 0.4962$ | $0.4657 \pm 0.4988$ | $+0.0271 \pm 0.3361 \uparrow$ |
| PIQA | $0.8580 \pm 0.3491$ | $0.8575 \pm 0.3496$ | $-0.0005 \pm 0.2523 \downarrow$ |
| OpenBookQA | $0.8680 \pm 0.3385$ | $0.8520 \pm 0.3551$ | $-0.0160 \pm 0.2603 \downarrow$ |
| ARC-Challenge | $0.9010 \pm 0.2986$ | $0.8976 \pm 0.3032$ | $-0.0034 \pm 0.1703 \downarrow$ |
| *Panel B: Sentence-level Understanding* | | | |
| RTE | $0.8592 \pm 0.3478$ | $0.8917 \pm 0.3108$ | $+0.0325 \pm 0.2599 \uparrow$ |
| MRPC | $0.7108 \pm 0.4534$ | $0.7721 \pm 0.4195$ | $+0.0613 \pm 0.3411 \uparrow$ |
| SST-2 | $0.9243 \pm 0.2645$ | $0.9358 \pm 0.2451$ | $+0.0115 \pm 0.1432 \uparrow$ |
| QNLI | $0.8574 \pm 0.3497$ | $0.8666 \pm 0.3401$ | $+0.0092 \pm 0.2684 \uparrow$ |
| QQP | $0.7833 \pm 0.4120$ | $0.8223 \pm 0.3822$ | $+0.0390 \pm 0.2566 \uparrow$ |
| CoLA | $0.7699 \pm 0.4209$ | $0.7162 \pm 0.4508$ | $-0.0537 \pm 0.3112 \downarrow$ |
| MNLI | $0.8292 \pm 0.3763$ | $0.8579 \pm 0.3492$ | $+0.0286 \pm 0.2638 \uparrow$ |
| WiC | $0.6097 \pm 0.4878$ | $0.6317 \pm 0.4824$ | $+0.0219 \pm 0.3257 \uparrow$ |

non-watermarked distributions.

### 5.3. Fidelity Evaluation

We evaluate the watermarked model on 15 benchmarks covering general reasoning and sentence understanding. As shown in Tab. 1, the watermarked model achieves performance comparable to the original baseline. On reasoning tasks, the degradation is marginal ($-0.0083$). On sentence-level tasks, we observe minor fluctuations, ranging from slight gains to small drops.

### 5.4. Ablation Studies

We investigate the effectiveness of individual components in Table 3. The results demonstrate that our proposed BiCoT framework achieves optimal detection performance. In the training formulation (Panel A), the watermark loss $L_{wm}$ is foundational; removing it naturally yields random-guess level performance. Crucially, in the detection phase (Panel B), the Re-anchoring mechanism proves vital: ablating this component causes a catastrophic drop in pAUC , validating its necessity for countering distribution drift. Furthermore, while the pruning strategy is not strictly required for high AUC, it significantly boosts the separation margin, ensuring robust detection boundaries in our main experiments.

### 5.5. Sensitivity Analysis

We analyze the sensitivity of our proposed method with respect to three key hyperparameters: the number of sentinels ($N$), the fraction of calibration data, and the noise level ($\sigma$). As shown in Figure 8, our approach demonstrates strong robustness across all settings. Specifically, the detection performance saturates rapidly with a small number of sentinels (Figure 8a) and maintains high AUC scores even with limited calibration data (Figure 8b). Furthermore, the

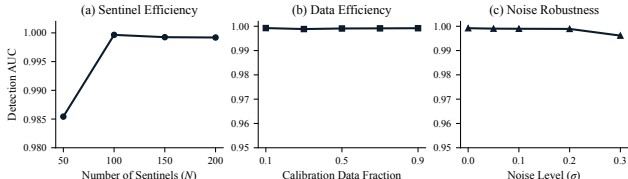

*Figure 8.* **Sensitivity Analysis.** We evaluate the robustness of the Detection AUC under three distinct settings: **(a) Sentinel Efficiency:** The performance improves with the number of sentinels ($N$) and saturates when $N > 100$, indicating that a small sentinel set is sufficient. **(b) Data Efficiency:** The method achieves competitive AUC even with a minimal fraction (0.1) of calibration data. **(c) Noise Robustness:** The detection capability is resilient to noise injection, maintaining high accuracy even as the noise level $\sigma$ increases to 0.3.

method exhibits negligible performance degradation against noise perturbations (Figure 8c), confirming its stability in practical scenarios.

### 5.6. Robustness Against Attacks

We evaluate the resilience of our watermarking scheme under a diverse set of perturbations, ranging from Gaussian noise to advanced adaptation techniques. Table 4 summarizes the detection metrics. We observe that the proposed method is highly robust to common perturbations. Specifically, under Gaussian noise ($\sigma = 0.05$) and full-parameter Supervised Fine-Tuning (SFT), the method maintains near-perfect detection performance (AUC $\approx 1.0$) and retains a Weighted Success Rate (WSR) of 100% at 1% FPR. Furthermore, the robustness generalizes to parameter-efficient tuning and compression scenarios. As shown in the bottom panel of Table 4, the watermark survives PEFT (LoRA) without degradation and maintains high stability even under aggressive quantization (QT) and other attack methods.

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

high dimensions), while watermarked traces align perfectly with the secret key.

**Attention Redistribution.** We further investigate the internal mechanism by analyzing the attention flow from non-digit to digit tokens. As shown in Figure 7, BiCoT exhibits a 19.1% increase in global attention towards numerical tokens compared to the base model. Notably, this global shift significantly outweighs the local attention boost (+9.4% within a 5-token window), indicating that the model learns to aggregate numerical information across long-range contexts rather than local proximity.

# 6. Conclusion

In our paper, BiCoT redefines the security boundaries of Large Language Models by shifting the watermarking frontier from observable outputs to the underlying Chain-of-Thought manifold. By identifying and anchoring the watermark within the high-saliency tokens that drive causal reasoning, we move beyond fragile surface-level perturbations toward a state of functional entanglement. Our framework demonstrates that ownership verification can be woven into the very fabric of model utility, creating a defense mechanism that is as indispensable as the reasoning logic itself. This paradigm not only ensures resilience against sophisticated adaptive attacks and distributional shifts but also establishes a theoretically grounded path for securing intellectual property in the increasingly complex landscape of generative AI. Beyond technical robustness, BiCoT offers a critical safeguard for the ethical deployment of AI, providing a reliable mechanism for traceability and accountability that mitigates the risks of model plagiarism and unauthorized redistribution in the global AI ecosystem.

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

# A. Proof of Theorem 3: Robust Type-I Error Control

In this section, we provide a formal derivation for **Theorem 3**. We establish that the Robust Subspace Registration (RSR) algorithm induces a test statistic that is *location-invariant* with respect to the underlying manifold drift. Consequently, we prove that the False Positive Rate (FPR) is strictly bounded by the significance level $\alpha$, regardless of the magnitude of the attack vector $\|\mathbf{d}\|$.

## A.1. Formal Setup and Definitions

Let the latent space be $\mathcal{X} \subseteq \mathbb{R}^V$. We consider a query sample $\mathbf{z} \in \mathcal{X}$ and a set of sentinel tokens $\mathcal{S} = \{\mathbf{s}_1, \ldots, \mathbf{s}_K\} \subset \mathcal{X}$.

**Hypothesis Definition.**

- **Null Hypothesis ($\mathcal{H}_0$):** The model is unwatermarked. The clean logits for both query and sentinels are independent and identically distributed (i.i.d.) drawn from a centered isotropic Gaussian background process:

$$\mathbf{z}_{clean}, \mathbf{s}_{k,clean} \sim \mathcal{N}(\mathbf{0}, \sigma^2 \mathbf{I}). \tag{8}$$

- **Drift Attack Model:** An adversary applies an affine transformation to the manifold. For any input, the observed logits are displaced by a constant drift vector $\mathbf{d} \in \mathbb{R}^V$:

$$\mathbf{z}_{obs} = \mathbf{z}_{clean} + \mathbf{d}, \quad \mathbf{s}_{k,obs} = \mathbf{s}_{k,clean} + \mathbf{d}. \tag{9}$$

**Estimator Definition.** Let $\hat{\boldsymbol{\mu}} : \mathbb{R}^{K \times V} \to \mathbb{R}^V$ be the robust location estimator. We employ the dimension-wise median operator over the sentinel set $\mathcal{S}_{obs}$. For the $j$-th dimension:

$$\hat{\mu}_j(\mathcal{S}_{obs}) = \mathrm{median}(\{s_{k,obs}^{(j)}\}_{k=1}^K). \tag{10}$$

## A.2. Invariance Lemma

We first establish the translation equivariance property of the estimator and the resulting invariance of the calibrated representation.

**Lemma A.1** (Affine Equivariance of the Median)**.** *For any drift vector* $\mathbf{d} \in \mathbb{R}^V$*, the estimator satisfies* $\hat{\boldsymbol{\mu}}(\mathcal{S}_{obs}) = \hat{\boldsymbol{\mu}}(\mathcal{S}_{clean}) + \mathbf{d}$.

*Proof.* Consider the $j$-th dimension. The observed set is $\{s_{k,clean}^{(j)} + d_j\}_{k=1}^K$. Since the median is a location-equivariant statistic (i.e., $\mathrm{med}(X + c) = \mathrm{med}(X) + c$ for scalar $c$), we have:

$$\hat{\mu}_j(\mathcal{S}_{obs}) = \mathrm{median}(\{s_{k,clean}^{(j)} + d_j\}) = \mathrm{median}(\{s_{k,clean}^{(j)}\}) + d_j = \hat{\mu}_j(\mathcal{S}_{clean}) + d_j. \tag{11}$$

This holds for all dimensions $j \in \{1, \ldots, V\}$, proving the vector equality. $\square$

## A.3. Proof of Error Control

**Theorem 3 Statement.** *Under* $\mathcal{H}_0$*, for any significance level* $\alpha \in (0,1)$ *and corresponding critical value* $\tau_\alpha = \Phi^{-1}(1-\alpha)$*, the RSR test statistic* $Z_{RSR}$ *satisfies:*

$$\sup_{\mathbf{d} \in \mathbb{R}^V} \mathbb{P}(Z_{RSR}(\mathbf{z}_{obs}) > \tau_\alpha \mid \mathcal{H}_0) \leq \alpha. \tag{12}$$

*Proof.* The RSR test statistic is defined as the projection of the registered query onto the signature unit vector $\mathbf{v}$:

$$S(\mathbf{z}_{obs}) = \langle \mathbf{z}_{obs} - \hat{\boldsymbol{\mu}}(\mathcal{S}_{obs}), \mathbf{v} \rangle. \tag{13}$$

Substituting the drift model and applying Lemma A.1:

$$\begin{aligned} S(\mathbf{z}_{obs}) &= \langle (\mathbf{z}_{clean} + \mathbf{d}) - (\hat{\boldsymbol{\mu}}(\mathcal{S}_{clean}) + \mathbf{d}), \mathbf{v} \rangle \\ &= \langle \mathbf{z}_{clean} - \hat{\boldsymbol{\mu}}(\mathcal{S}_{clean}), \mathbf{v} \rangle \\ &= S(\mathbf{z}_{clean}). \end{aligned} \tag{14}$$

Equation (14) reveals that the statistic depends *exclusively* on the clean distribution $\mathcal{N}(\mathbf{0}, \sigma^2\mathbf{I})$, and is algebraically independent of $\mathbf{d}$.

We now analyze the distribution of $S(\mathbf{z}_{clean})$. Let $\mathbf{r} = \mathbf{z}_{clean} - \hat{\boldsymbol{\mu}}(\mathcal{S}_{clean})$. Since $\mathbf{z}_{clean}$ and $\mathcal{S}_{clean}$ are drawn from centered Gaussians, $\mathbf{r}$ is a zero-mean random vector. The projection of a Gaussian vector onto a fixed direction $\mathbf{v}$ remains Gaussian. Therefore, $S(\mathbf{z}_{clean}) \sim \mathcal{N}(0, \sigma_{eff}^2)$.

The standardized Z-score is given by $Z_{RSR} = S(\mathbf{z}_{clean})/\sigma_{eff}$. By definition, $Z_{RSR} \sim \mathcal{N}(0, 1)$. The probability of observing a False Positive is given by the tail mass of the standard normal distribution:

$$\mathbb{P}(Z_{RSR} > \tau_\alpha \mid \mathcal{H}_0) = 1 - \Phi(\tau_\alpha) = \alpha. \tag{15}$$

Since Eq. (14) holds for *any* $\mathbf{d}$, the supremum of this probability over the entire vector space $\mathbb{R}^V$ is exactly $\alpha$. The RSR detector is thus uniformly robust against affine drift attacks. $\qquad\square$

## B. Proof for Proposition 4.2

*Proof.* Substituting $\mathbf{z}_{obs}$ into Eq. (6), the term $\hat{\boldsymbol{\mu}}_{drift}$ cancels the attack-induced bias $\mathbf{W}_{head}\mathbf{d}_{drift}$. The score reduces to $\langle \mathbf{z}_{clean} + \boldsymbol{\epsilon}, \mathbf{W}_{head}\mathbf{v} \rangle$, which depends solely on the clean alignment and random noise, independent of the drift magnitude $\|\mathbf{d}_{drift}\|$. $\qquad\square$

## C. The settings of the Sec. 3.1

This appendix supplements the foundational analysis in Section 3 with formal metric definitions and experimental configurations.

This appendix provides the formal definitions, mathematical formulations, and detailed experimental configurations corresponding to the foundational analysis in Section 3.

### C.1. Metrics for Token Role Analysis (Sec. 3.1)

In Section 3.1, we distinguish the informational roles of Chain-of-Thought (CoT) and Final Answer tokens using three complementary metrics.

**Effective Capacity.** We quantify the information density of a generated sequence using the Effective Token Count. For a given sequence with an empirical token distribution $\hat{p}(v)$ over the vocabulary $\mathcal{V}$, the capacity $C$ is defined as the exponential of the Shannon entropy:

$$C = \exp\left( -\sum_{v \in \mathcal{V}} \hat{p}(v) \ln \hat{p}(v) \right). \tag{16}$$

A higher capacity indicates a high-entropy distribution, suggesting that the sequence occupies a broader representational space suitable for embedding auxiliary signals.

**Diversity.** We measure the semantic variance between independent generations $T_A$ and $T_B$ sampled from the same prompt. Treating each sequence as a bag of unique unigrams, we compute the Jaccard Similarity $J$:

$$J(T_A, T_B) = \frac{|T_A \cap T_B|}{|T_A \cup T_B|}. \tag{17}$$

We observe that CoT traces exhibit significantly lower Jaccard similarity compared to final answers, implying that the reasoning process possesses greater intrinsic diversity than the converged solution.

**Token Importance.** To assess the distribution of information, we measure the sensitivity of the ground truth output $y$ to random token masking. Let $x$ be the original context and $\tilde{x}_r$ be the context with a ratio $r$ of tokens masked. The importance is quantified by the drop in the log-probability of the correct answer:

$$\Delta\mathcal{L} = \log P(y \mid x) - \log P(y \mid \tilde{x}_r). \tag{18}$$

The experimental results show that $\Delta\mathcal{L}$ increases gradually for CoT tokens, indicating distributed importance, whereas it spikes sharply for final answer tokens, indicating a reliance on specific critical positions.

**C.2. Identification of Structural Anchors (Sec. 3.2)**

Section 3.2 posits that reasoning dynamics are governed by sparse "Structural Anchors." We identify these anchors using gradient saliency and validate them via perturbation stability.

**Gradient Saliency Analysis.** We define anchors as the subset of CoT tokens that exert maximal causal influence on the final answer. We compute the gradient of the cross-entropy loss of the *answer tokens* ($y_{ans}$), conditioned on the prompt and CoT ($y_{cot}$), with respect to the hidden states $h_t$ at layer $L - 4$. The saliency score $S_t$ for a token at position $t$ is the $L_2$ norm of this gradient:

$$S_t = \|\nabla_{h_t} \mathcal{L}(y_{ans} \mid x, y_{cot})\|_2. \tag{19}$$

Our analysis reveals that $S_t$ follows a heavy-tailed distribution, with mass concentrated on numerical tokens (digits) in mathematical reasoning tasks.

**Stability Threshold.** To determine the robustness of these anchors, we measure the Exact Match (EM) rate of the final answer under perturbation. We define the stable operating regime as the range of injection strengths $\alpha$ where the EM rate exceeds a strict reliability threshold $\tau$:

$$\text{Stability}(\alpha) = \mathbb{E}_x \left[ \mathbb{I}(\text{EM}(y_\alpha, y_{GT})) \right] \geq 0.85. \tag{20}$$

This confirms that while anchors are causally significant, they possess a tolerance margin that allows for watermark embedding.

**C.3. Signal Propagation Dynamics (Sec. 3.3)**

Section 3.3 analyzes the linearity of signal transmission from the CoT internal representations to the final output logits.

**Perturbation Mechanism.** We inject additive interference into the hidden states of the CoT at layer $l = L - 4$. Let $h \in \mathbb{R}^d$ be the original hidden state. The perturbed state $\tilde{h}$ is computed as:

$$\tilde{h} = h + \alpha \cdot \mathbf{v}, \tag{21}$$

where $\alpha$ is the scalar injection strength and $\mathbf{v} \sim \mathcal{N}(0, I)$ is a random noise vector normalized to unit norm.

**Linearity of Propagation.** We quantify the signal propagation by measuring the shift in the output probability distribution. Specifically, we calculate the $L_2$ distance between the logits of the original and perturbed models:

$$\delta(\alpha) = \|\text{Logits}(\tilde{h}) - \text{Logits}(h)\|_2. \tag{22}$$

To verify the linearity of this propagation, we compute the Pearson correlation coefficient $r$ between the injection strength $\alpha$ and the induced shift $\delta(\alpha)$ across the probing range $\alpha \in [0, 0.05]$. High correlation values ($r \approx 1.0$) confirm that the CoT acts as a linear channel for transmitting the watermark signal before saturation occurs.

# D. Baseline Discussion

Existing model watermarking techniques for Large Language Models (LLMs) can be broadly categorized into white-box and black-box paradigms, distinguished primarily by their verification mechanisms.

**D.1. White-box Watermarking**

White-box watermarking operates under the assumption that the verifier has full access to the model's parameters. These approaches typically embed ownership signatures directly into the model weights (Liang et al., 2026). Verification is conducted by inspecting the internal parameters to extract the embedded signal. While these methods allow for robust embedding, the requirement for white-box access limits their utility in scenarios where models are deployed as black-box APIs or encapsulated services.

**D.2. Black-box Watermarking**

In contrast, black-box watermarking verifies ownership solely through API queries, without requiring access to the model's internal weights (Yamabe et al., 2025; Xiong et al., 2025). Current literature in this domain is dominated by three primary strategies: backdoor-based methods, model editing-based methods, and fingerprinting approaches.

**Backdoor-based Approaches.** This stream of research focuses on injecting specific trigger-action patterns into the model (Shao et al., 2024; Li et al., 2023). The core idea is to force the model to output a specific watermark signal when a predefined trigger appears in the input. For instance, Instructional Fingerprinting (IF) methods (Xu et al., 2024) implant a private key as an instruction backdoor, causing the model to generate specific text when the key is present. This can be achieved through full-parameter supervision (IF-sft) or lightweight embedding updates (IF-emb). However, forcing the model to overfit to a small set of fixed triggers introduces significant drawbacks. It fundamentally compromises *fidelity* on complex tasks and degrades *generalization* due to interference from the backdoor mechanism (Gu et al., 2017; Hu et al., 2024; Puah et al., 2024). Additionally, the reliance on fixed patterns renders these watermarks susceptible to detection and filtering, and verification becomes unreliable if triggers are leaked or spoofed (Wang et al., 2019b; Liu et al., 2019).

**Model Editing-based Approaches.** To mitigate the inefficiencies of full fine-tuning used in some backdoor methods, model editing-based approaches leverage knowledge updating algorithms to inject watermarks as specific facts (Li et al., 2025b; Yue et al., 2025). Techniques such as *met* (Gao et al., 2024) and SEAL (Dai et al., 2025) fall into this category. Specifically, SEAL proposes a subspace-anchored watermarking framework that utilizes model editing to align the latent representations of anchor samples with orthogonal bit vectors. By integrating the watermark into the model's knowledge space rather than creating behavioral outliers, these methods improve stealthiness and efficiency. Nevertheless, this paradigm often treats the watermark as static knowledge to be memorized, which can be fragile against distribution shifts or adaptive attacks on prompt structures (Cheng et al., 2023; Wang et al., 2024c).

**Fingerprinting and Active Verification.** Distinct from injection-based watermarking, fingerprinting methods focus on identifying unique model characteristics or injecting identifiable features for ownership verification. *llmmap* (Pasquini et al., 2025) employs an active fingerprinting approach, sending carefully crafted queries to an API and analyzing the response distribution to identify the specific model version without modifying its weights. Conversely, iSeal (Xiong et al., 2025) introduces an encrypted fingerprinting mechanism designed for scenarios where the adversary controls the inference process. It injects unique features into the model and an external module, utilizing error correction to ensure reliable verification even under unlearning or response manipulation attacks.

Notably, the IF does not perform well in our main experiment, which is just the same as the results in SEAL (Dai et al., 2025).

# E. More Discussion about the Experiments

In this section, we provide detailed clarifications regarding the feasibility of the black-box protocol, the robustness of the subspace registration mechanism, and the generalization capabilities of the anchor-based watermarking framework.