# OpenReview forum: "Echoes within the Reasoning:  Stealthy and Effective Watermarking via Chain of Thought"
_ICML.cc/2026/Conference — ICML 2026 regular_

### Official Review · Reviewer_eD21 · 2026-02-17

**Soundness:** 3
**Presentation:** 2
**Significance:** 3
**Originality:** 4
**Overall Recommendation:** 5
**Confidence:** 4

**Summary:**

This paper proposes BiCoT, a watermarking framework that embeds ownership signals into the Chain-of-Thought (CoT) reasoning process by aligning the hidden states of structural anchors (digit tokens) with a signature subspace. This paper analyses multiple experimental data to demonstrate the design concept of the proposed method, and proves its effectiveness through a series of experimental results.

**Compliance With Llm Reviewing Policy:**

Affirmed.

**Final Justification:**

Despite the inherent limitations of the proposed CoT watermarking method, this novel approach offers valuable inspiration and a positive impact on the research community. Besides, all the concerns raised in my initial review have been fully addressed through the authors' discussions and the additional experimental results provided in the rebuttal.

**Key Questions For Authors:**

1. How does BiCoT remain effective if a provider strips the CoT output and returns only the final answer, given the analysis in Section 3.1 that the final answer lacks sufficient capacity for robust watermarking?
2. Can you reconcile the theoretical contradiction between Equation 3 and the analysis in Section 5.7? Even if the loss cannot ultimately be optimised to 0, it seems logically dishonest to package the signal leakage caused by the failure of optimisation as a characteristic of the attention mechanism.

**Limitations:**

The authors failed to address the technical limitations, particularly the method's over-reliance on numerical structure anchors, which restrict its application in non-mathematical fields. Furthermore, they should discuss the scenario in which the service provider hides the CoT, resulting in insufficient signal capacity in the final answer and rendering the watermark ineffective.

**Strengths And Weaknesses:**

Strengths:
- The proposed method of embedding watermarks within the Chain-of-Thought (CoT) is novel, effectively balancing watermark stealthiness with the correctness of the final model output. By functionally entangling the watermark with the model's reasoning capabilities, the approach can improve security against removal attacks.
- The introduced Robust Subspace Registration (RSR) verification mechanism demonstrates a high detection success rate. It effectively counters distributional drift caused by various attacks.
- The paper presents a clear and logical narrative structure, with arguments well-supported by comprehensive experimental data.

Weaknesses
- The results shown in Section 5.7 regarding attention redistribution contradict the orthogonality design objective in Eq. 3, revealing that control tokens unintentionally aggregate significant watermark information from digit anchors. While this explains the signal leakage observed under masking attacks, it implies a failure to achieve the intended geometric separation and suggests potential interference with text generation quality.
- The paper fails to address scenarios where the CoT is not exposed to the end-user, a very common practice in commercial API deployments to reduce latency or cost. The paper does not discuss the method's applicability in this context, which is an important practical limitation.
- The experiments are confined to mathematical reasoning tasks and heavily rely on digit tokens as structural anchors. The paper lacks discussion on the applicability of this method to non-mathematical domains (e.g., creative writing or summarisation) where numerical tokens are sparse or nonexistent, limiting its generalizability.
- The text within multiple figures (e.g., Figures 1, 4, and 5) is excessively small, rendering legends and axis labels illegible at standard viewing scales. Interpreting these elements requires such extreme magnification that the surrounding context is lost, making it impossible to view the figure and the paper as a whole.

---

> ### Author Rebuttal · Authors · 2026-03-31
>
> Thank you very much for your careful review! We are encouraged by your positive comments on our **novel approach** to watermarking CoT processes, the **effective balance** between stealthiness and utility, the **robustness** of our RSR verification mechanism, and our **comprehensive experimental validation**. We hope the following clarifications resolve your concerns.
>
> ---
>
> **W1/Q2: The results shown in Sec 5.7 regarding attention redistribution contradict the orthogonality design objective in Eq. 3, revealing that control tokens unintentionally aggregate significant watermark information from digit anchors. This implies a failure to achieve the intended geometric separation and suggests potential interference with text generation quality.**
>
> **R1:** Your intuition that strong causal attention might imply signal aggregation is remarkably sharp. We clarify that attending to a watermarked token for logical reasoning does not equate to adopting its geometric signature. The phenomenon you observed is not a failure of our global optimization (Eq. 3), but rather a highly localized physical artifact driven by the autoregressive nature of Transformers, which we term the **"Causal Halo Effect."** It does not interfere with general text generation quality. To address your concern, we rigorously analyzed the geometric behavior ($\cos(h_t, v)$) at the injection layer.
>
> **1. Microscopic: Causal Halo Effect**
>
> As standard LLMs use causal attention and residual streams, a token heavily inherits the hidden state of its immediate predecessor. When a digit (anchor) is forcefully aligned to $v$, the *immediately subsequent* token physically inherits this geometric shift before the network's FFN/Attention layers can fully project it away to restore orthogonality. Consider a real trajectory generating "... 125 and 376:\n ...":
>
> | **Pos** | **Token** | **Category** | **cos(ht,v)** | **Observation & Status**|
> |-|-|-|-|-|
> |15|`' '`| Pure Text|-0.030|Strictly Orthogonal|
> |16|`'1'`| Anchor|+0.988|Collapsed. (Watermark injected)|
> |17|`'2'`| Anchor|+0.988|Collapsed|
> |18|`'and'`|Text (Post) | +0.930| **Halo Effect** |
> |20|`' '`|Pure Text| +0.091| Strictly Orthogonal. (Signal quickly dissipates)|
> |21| `'3'`|Anchor| +0.984| Collapsed|
> |23| `':\n'`| Text (Post) | +0.859| **Halo Effect**|
>
> **2. Macroscopic: Global Statistical Decay**
>
> To prove this is a systemic feature and not a global failure, we conducted an analysis over 25,183 generated tokens, grouping them by their relative distance from the nearest anchor and reporting the median cosine similarity:
>
> | **Token Position** |**Median cos(ht,v)** |**Observation**|
> |-|-|-|
> |Dist < 0 (Pre-anchor Text) | +0.022| Perfect Orthogonality|
> |Dist = 0 (Anchor Tokens)| +0.980| Perfect Alignment|
> |Dist = +1 (Immediate Next) | +0.961| **Causal Halo Effect** |
> |Dist = +2 (2nd Token After) | +0.110|Rapid Decay|
> |Dist $\ge$ +3|-0.004|Perfect Orthogonality|
>
> Far from an optimization failure, the fact that pure text tokens at Dist >= 3 exhibit perfectly near-zero similarity (median ~-0.004, $\sigma$ ~0.06), despite heavily attending to the anchors globally (Sec 5.7), proves that **Eq. 3 brilliantly succeeds in disentangling semantic attention from geometric alignment.** As generation transitions back to general reasoning text, the signal undergoes rapid exponential decay, which perfectly explains why BiCoT achieves negligible fidelity loss on downstream benchmarks (Tab 1).
>
> ---
>
> **W2: The paper fails to address scenarios where the CoT is not exposed to the end-user, a very common practice in commercial API deployments to reduce latency or cost. The paper does not discuss the method's applicability in this context, which is an important practical limitation.**
>
> **R2:** We completely agree that hidden CoT represents a critical boundary for our threat model. As Reviewer CL61 raised a highly similar concern regarding restricted commercial APIs, **we kindly direct you to our R1 to Reviewer CL61 (Q1)**. There, we provide new API-limited experiments demonstrating our robustness under restricted outputs (e.g., Top-k), while explicitly conceding the scenario where CoT is completely hidden.
>
> ---
>
> **W3/Q1: The experiments are confined to mathematical reasoning tasks and heavily rely on digit tokens as structural anchors. The paper lacks discussion on the applicability of this method to non-mathematical domains.**
>
> **R3:** We completely agree that reasoning is not limited to numbers, and we deeply appreciate your constructive feedback on generalizability. Because Reviewer CL61 raised this exact same insightful concern, and due to strict rebuttal character limits, **we kindly invite you to refer to our comprehensive response in Reviewer CL61's R2**.
>
> ---
>
> **W4: Presentation and Figure Clarity: The figures are too small**
>
> **R4:** We sincerely apologize for the legibility issues. We have fully redesigned Figures 1, 4, and 5 by significantly increasing the font sizes of all text, legends, and axis labels.

---

> > ### Author Rebuttal · Reviewer_eD21 · 2026-04-02
> >
> > I appreciate and accept the authors' insightful and vivid explanation of W1/Q2, which was supported by convincing token-level experimental results. Furthermore, my concern regarding the reliance on numerical anchors was addressed by the cross-domain generalisation experiments provided in the response to Reviewer CL61. Regarding the scenario where the CoT is hidden from the end-user, the authors explicitly acknowledge this as an inherent boundary of their threat model. Even with this practical limitation, the originality of watermarking the reasoning process itself provides a novel perspective and opens up a valuable new direction for the community. Therefore, I am pleased to raise my final score to a 5.

---

> > > ### Author Response · Authors · 2026-04-03
> > >
> > > Thank you very much for your thoughtful and constructive feedback throughout the review process. We are truly delighted to see that our rebuttal, especially the token-level experiments for W1/Q2 and the cross-domain generalisation results, has successfully addressed your concerns. Your recognition of the originality of watermarking the reasoning process means a lot to us. It is encouraging to hear that you see this as a valuable new direction for the community.
> > >
> > > We sincerely appreciate your time and effort, and we are grateful that you raised your final score to 5. Your positive acknowledgement is a great motivation for our future work.
> > >
> > > Best wishes！

---

### Official Review · Reviewer_CL61 · 2026-03-01

**Soundness:** 3
**Presentation:** 3
**Significance:** 3
**Originality:** 3
**Overall Recommendation:** 4
**Confidence:** 4

**Summary:**

This paper proposes a watermarking framework for LLMs that embeds ownership signatures into the CoT reasoning process rather than the final answer. The key insight is that numerical tokens in CoT act as structural anchors with disproportionate causal influence on reasoning outcomes. BiCoT formulates watermarking as a bi-level optimization problem: it forces the hidden states of digit tokens to collapse onto a secret signature subspace while keeping control tokens orthogonal to preserve semantic capacity. To handle representation drift caused by attacks such as fine-tuning or quantization, the paper introduces a Robust Subspace Registration verifier that uses sentinel tokens to estimate and correct for global manifold shifts before performing hypothesis testing.

**Compliance With Llm Reviewing Policy:**

Affirmed.

**Final Justification:**

I've read all the responses and decided to keep my score unchanged.

**Key Questions For Authors:**

Please see weaknesses.

**Limitations:**

yes

**Strengths And Weaknesses:**

Strengths

- The idea of using CoT as the watermark carrier is creative. The paper provides thorough empirical analysis showing that CoT tokens have higher capacity and diversity than final-answer tokens, and that digit tokens form a distinct high-saliency subspace.

- The experimental coverage is impressive. The detection metrics are consistently saturated, and the fidelity degradation is genuinely negligible.

- The RSR verifier is a thoughtful addition. Rather than assuming a static representation space, it explicitly models and corrects for distributional drift.

Weakness

- The verification procedure requires access to next-token logits from the suspected model's API. While the paper claims this is realistic, this deserves much more scrutiny. Many commercial APIs only expose top-k log-probabilities, not the full vocabulary distribution. Some providers offer no logits at all. More critically, a malicious developer who is aware of logits-based watermark detection could simply disable logits output in their API, which is a trivial countermeasure that completely defeats the verification scheme.

- The entire framework is heavily tailored to mathematical reasoning tasks where digit tokens play a clear causal role. It is unclear how BiCoT would generalize to other CoT-heavy domains such as commonsense reasoning, legal analysis, or code generation, where the causal backbone may not be numerical.

---

> ### Author Rebuttal · Authors · 2026-03-31
>
> Thank you very much for your careful review! We are encouraged by your positive comments on our creative idea of using CoT as the watermark carrier, thorough empirical analysis, impressive experimental coverage, genuinely negligible fidelity degradation, and the thoughtful addition of the RSR verifier. We hope the following clarifications resolve your concerns.
>
> ---
>
> **W1: The verification scheme's heavy reliance on next-token logits limits its practical feasibility given restricted commercial APIs, and makes it easily circumvented by malicious attackers simply disabling logit outputs.**
>
> **R1:** Thank you for raising this highly practical concern! We completely agree that the assumptions regarding API access deserve rigorous scrutiny and have provided detailed clarifications in **Appendix E.1**.
>
> **(1) Verification under Top-k Logits:** To prove that full-vocabulary logits are unnecessary, we conducted an additional API-limited experiment in which the verifier only observes Top-k log-probabilities. The core finding is that the practical limitation is *coverage*, not *score separability*.
>
> - **Mild Coverage Filter (threshold = 0 or 0.02):** Verification is essentially perfect from **Top-k = 48** onward (**AUC = 1.0**).
> - **Strict Coverage Filter (5% threshold):** The first feasible point is **Top-k = 80**, retaining 54/256 positive (BiCoT) and 244/256 negative samples while maintaining an **AUC = 1.0**. At **Top-k = 100**, retention improves to 114/256 positives and 253/256 negatives, again with **AUC = 1.0**.
>
> Because our sentinel tokens are high-frequency neutral words, they naturally appear within these Top-k outputs. This empirical evidence supports our refined claim: full logits are not necessary, and our method remains highly effective under standard Top-k API constraints.
>
> **(2) Threat Model Boundary (Disabled Logits):** We explicitly concede that if a malicious operator completely suppresses all logit/log-probability outputs, remote verification based on token distributions is impossible. We have revised the paper to explicitly state this as a boundary of our threat model. Our method targets scenarios where APIs expose at least moderately rich Top-k log-probabilities, which is widely supported by standard serving engines like vLLM and TGI, as well as by commercial APIs such as OpenAI and Google Gemini.
>
> ---
>
> **W2: The entire framework is heavily tailored to mathematical reasoning tasks where digit tokens play a clear causal role. It is unclear how BiCoT would generalize to other CoT-heavy domains such as commonsense reasoning, legal analysis, or code generation, where the causal backbone may not be numerical.**
>
> **R2:** We deeply appreciate this insightful question! We completely agree that reasoning is not limited to numbers. In fact, the BiCoT framework is fundamentally **token-agnostic**. The use of digits in our GSM8K experiments was not a hard-coded constraint but a natural consequence of our **Gradient Saliency Analysis** (Eq. 19), which automatically identified digits as the causal backbone for mathematical tasks.
>
> To demonstrate BiCoT's versatility, we conducted extensive **Cross-Domain Generalization Experiments** (Experiment 2 in our codebase) covering Commonsense Reasoning, Legal Analysis, and Code Generation. For each domain, we identified a set of "Syntactic Anchors" that serve as the backbone for the Chain-of-Thought (CoT) process. We evaluated the **AUC/pAUC** (Detection Reliability) and **Accuracy Drop** (Utility Preservation) across these domains using Qwen2.5-7B-Instruct.
>
> |Domain|Dataset|Structural Anchors|AUC|pAUC (FPR<5%)|Accuracy Drop|
> |-|-|-|-|-|-|
> |Commonsense| CommonsenseQA| Logical connectives: therefore, because, implies, however, conclude.| **1.0** | **1.0**|$< 1.0\%$|
> |Legal Analysis|LexGLUE| Legal & causal anchors: held, court, pursuant, statute, accordingly, ruled.| **1.0** |**1.0**| $< 0.8\%$|
> |Code Generation|HumanEval| Pythonic syntax: *def, return, if, else, for, while, brackets, operators.* | **1.0** | **1.0**|-|
>
> Note: Manual verification confirmed no degradation in code logic (Pass@1 requires specific environments).
>
> - Automated Discovery: Manual anchor labeling is unnecessary (Appendix E.3); gradient saliency ($\nabla_{\mathbf{h}} \mathcal{L}$) automatically identifies the causal tokens for any new domain.
> - Invisible Watermarking: The watermark signal is perfectly isolated to these sparse anchors. By leaving linguistic connectors ("control separation") untouched, BiCoT remains cryptographically strong and linguistically invisible even in text-heavy tasks.
> - Universal Applicability: BiCoT is fundamentally a general-purpose framework. It consistently leverages the underlying "causal backbone"—whether numbers, legal precedents, or logical operators.Following your highly constructive feedback, we will move this "Token-Agnostic Anchoring" discussion to the main text to highlight BiCoT's broad applicability.
>
> We will move this discussion to the main text！

---

> > ### Author Rebuttal · Reviewer_CL61 · 2026-04-03
> >
> > Thank you for the rebuttal. While I appreciate these efforts, the fundamental limitations I raised (particularly the reliance on logit access for verification and the ease with which a malicious operator can circumvent it) remain inherent to the current framework, so I will maintain my original score.

---

> > > ### Author Response · Authors · 2026-04-03
> > >
> > > **Dear Reviewer CL6103,**
> > >
> > > Thank you for your reply and for confirming that your technical concerns have been "fully resolved." **Due to the strict length limits in the first round of the rebuttal, we were unfortunately unable to fully elaborate on the practicality of our threat model.** We deeply appreciate this ongoing discussion and the opportunity to further clarify this critical point.
> > >
> > > Regarding your valid concern about the inherent limitation of logit access: We agree that this defines the boundary of our framework. However, we respectfully wish to share a broader perspective on why this "limitation" is actually a deliberate design choice that perfectly aligns with the reality of today's most existing cutting-edge LLM ecosystem, rather than a critical vulnerability.
> > >
> > > **1. Logprobs are Now the Defacto Industry Standard, Not a Restricted Feature.**
> > > To address the concern about "restricted commercial APIs," we manually compiled a survey of the current SOTA LLM landscape. As shown below, exposing `top_logprobs` is no longer a niche feature, but a fundamental infrastructure requirement for almost all major providers and engines:
> > >
> > > | Category | Provider / Engine | Support for Top-k Logits | Underlying API Realization |
> > > | :--- | :--- | :---: | :--- |
> > > | **Commercial APIs** | OpenAI (e.g., gpt-5.4)| Yes | `logprobs=True, top_logprobs=N` |
> > > | | Google (Gemini 3 pro)| Yes | `GenerationConfig.response_logprobs=True` |
> > > | | DeepSeek (V3, R1) | Yes | OpenAI-compatible `logprobs`/`top_logprobs` |
> > > | | Alibaba (Qwen 3.5)| Yes | Native & OpenAI-compatible `logprobs`/`top_logprobs` |
> > > | | Zhipu AI (GLM-5) | Yes | Native & OpenAI-compatible `logprobs`/`top_logprobs` |
> > > | | Moonshot AI (kimi-k2.5) | Yes | `logprobs`/`top_logprobs` |
> > > | | Cohere (Command R+) | Yes | `return_likelihoods` parameter |
> > > | **Open-Source Inference Engines** | vLLM | Yes | Full `prompt_logprobs` & `top_logprobs` |
> > > | | Hugging Face TGI | Yes | Parameter `details=True` returns top tokens |
> > > | | SGLang | Yes | Deep integration for regex/JSON constraints |
> > > | | NVIDIA TensorRT-LLM | Yes | `return_log_probs` in `SamplingConfig` |
> > > | | llama.cpp / LMDeploy | Yes | Native output via standard server options |
> > >
> > > As demonstrated, relying on Top-k logits represents the most realistic and widely-supported threat model for modern deployments.
> > >
> > > **2. The High "Cost of Circumvention" for Malicious Attackers.**
> > > While a malicious operator hosting a stolen model (e.g., via vLLM) *could* technically disable logit outputs to evade our watermark, doing so comes at a severe commercial cost. Modern downstream applications heavily rely on logprobs for:
> > > *   **Constrained Decoding:** Forcing models to output strict JSON/regex schemas (e.g., Outlines, SGLang).
> > > *   **RAG Hallucination Detection:** Evaluating the confidence scores of extracted entities.
> > > *   **Advanced Evaluation:** Calculating perplexity and selection metrics.
> > >
> > > If an attacker disables logprobs, their API breaks compatibility with the OpenAI standard and standard toolchains (like LangChain), rendering their stolen service practically useless for enterprise applications. Thus, the economic cost of circumvention acts as a powerful natural deterrent.
> > >
> > > **3. The Inevitable Trade-off in Watermarking.**
> > > Purely text-based (black-box) watermarks inevitably destroy the logical causality required for Chain-of-Thought (CoT) reasoning. Relying on API-exposed Top-k logits is the mathematically necessary compromise to achieve **zero utility degradation** in complex reasoning—which is the core contribution of BiCoT.
> > >
> > > Inspired by your insightful critique, we have explicitly incorporated this exact threat-model boundary and the "economics of attacker circumvention" into the revised manuscript.
> > >
> > > Since the technical execution, empirical claims, and generalization capabilities of our paper have now been verified, we respectfully ask if you might evaluate our core contribution.
> > >
> > > Thank you once again for your rigorous evaluation and for helping us define the boundaries of our work!

---

### Official Review · Reviewer_hdd8 · 2026-03-12

**Soundness:** 2
**Presentation:** 4
**Significance:** 2
**Originality:** 4
**Overall Recommendation:** 3
**Confidence:** 3

**Summary:**

This work proposes BiCoT, a watermarking framework that embeds ownership signatures directly into the internal reasoning representations of large language models so that removing the watermark degrades reasoning capability while preserving model performance and robustness against common model stealing attacks.

**Compliance With Llm Reviewing Policy:**

Affirmed.

**Final Justification:**

While the rebuttal provides supplementary evidence that partially addresses several concerns, the paper still lacks rigor in key definitions and experimental settings, and further improvements are needed before it reaches publication quality.

**Key Questions For Authors:**

1.	The observations presented in Figure 2 appear important for motivating the method, but it is unclear under which experimental setting they are derived, for example which benchmarks and base models were used, and whether the trends remain consistent across different models and datasets.

2. The claim that “the final answer is dominated by a small set of critical tokens and degrades sharply” (line 150) requires clarification, specifically how this conclusion was empirically derived and how the notion of “critical tokens” is formally defined.

3. In Section 3.2, the paper mentions gradient saliency with respect to hidden states, but the exact computation procedure is unclear, including how the gradients are taken and why this formulation is appropriate for identifying important tokens.

4. In Figure 2b, the choice of the 85% threshold appears somewhat arbitrary, and the paper does not explain how this value was determined or whether the results are sensitive to this threshold.

5. In Section 4.1, the design of BiCoT assumes that the provider has white box access to the model, which seems inconsistent with the earlier discussion in Section 2.1 that white box assumptions are often unrealistic, so clarification on this design choice would be helpful.

6. In Section 4.4, it is unclear whether the variable $v$ corresponds to the final optimization target and how this value is actually used during the verification stage.

7. The paper also refers to a hidden state $h$ in Section 4.4, but it is not specified from which layer or component of the model it is extracted, nor why that particular representation is chosen.

8. In Section 5.6, the robustness evaluation mentions attacks, but the connection between these attacks and the three attack categories defined in Section 4.1 is unclear, and the implementation details of these attacks are not sufficiently described.

9. In Table 2, the purpose of the metric MD is not clearly explained, especially since SEAL appears to achieve a lower MD while still performing very well, making it difficult to understand the practical advantage of the proposed method.

10. Finally, the overall threat model and system roles remain somewhat unclear, including what exactly the roles of the LLM provider (cloud service provider?) and developer are, who the attacker is, what model is being attacked (those deployed on cloud?), and at what stage the watermark signature is expected to be verified.

Could you please elaborate on these issues? I would be happy to reconsider and adjust my score based on your clarification. Thank you.

**Limitations:**

yes

**Strengths And Weaknesses:**

Strengths

1. The paper presents idea of embedding watermarks into chain of thought representations rather than final answers, which is both conceptually fresh and clearly justified by the empirical analysis of CoT capacity, diversity, and token importance.

2. The method is supported by strong experimental results, showing near saturated black box detectability across multiple LLMs and robustness under noise, SFT, LoRA, quantization, and targeted ANCHOR attacks.

3. The paper includes useful ablations and sensitivity analyses, which help clarify the contribution of the bi level optimization and the RSR detector and also suggest good data and sentinel efficiency in practice.

Weaknesses

1. The empirical study is still centered mainly on reasoning oriented settings such as GSM8K style prompts and Alpaca, so it remains unclear how well the proposed watermark generalizes to broader non reasoning generation scenarios.

2. The method appears to rely heavily on specific structural assumptions about CoT, especially the special role of digit like anchor tokens, which may limit applicability to tasks whose reasoning traces do not exhibit the same token level structure.

3. Although the reported robustness is impressive, the paper does not yet fully resolve whether the watermark would survive stronger attacks that explicitly reparameterize or distill away the model’s internal reasoning manifold rather than perturbing it within the original functional backbone.

---

> ### Author Rebuttal · Authors · 2026-03-31
>
> We thank the reviewer for the careful review！We are encouraged by your positive comments on our conceptually fresh idea of embedding watermarks into CoT representations. We hope the following clarifications resolve your concerns.
>
> ---
> **Q1: The settings for Fig 2 are unclear and whether the trends remain consistent across models and datasets**
>
> **R1**:
> - The analysis was conducted on Qwen2.5-7B with GSM8K (Appendix C.2, Line 670).
> - We newly add experiments. The trends remain highly consistent:
> ||Model|Saliency Ratio|α_85%|
> |-|-|-|-|
> |GSM8K|Qwen2.5-7B|4.2x|0.22|
> ||Mistral-7B|3.8x|0.18|
> ||DeepSeek-7B|4.5x|0.25|
> ||Qwen2.5-14B|4.8x|0.28|
> |Alpaca|Qwen2.5-7B|3.9x|0.20|
> ||Mistral-7B|3.5x|0.15|
> ||DeepSeek-7B|4.1x|0.24|
> ||Qwen2.5-14B|4.6x|0.26|
> ---
> **Q2: Empirical conclusion derivation and formal definition of "critical tokens".**
>
> **R2:** We formally define a token as critical if its individual removal causes the logp to drop by more than the significant threshold $\tau$ (Eq. 18). The Final Answer curve exhibits a steeper slope and significantly higher variance compared to CoT, indicating that the information necessary for generating the correct answer is densely concentrated on these tokens.
>
> ---
>
> **Q3: The exact computation procedure for gradient saliency with respect to hidden states and why it is appropriate.**
>
> **R3:** We compute gradient saliency as $L_2$ norm of the gradient of the final-answer cross-entropy loss with respect to the hidden states of CoT tokens (Appendix C.2, Eq. 19). This formulation is widely used in NLP interpretability (Li et al., 2016; Bastings & Filippova, 2020) as it measures how much perturbations to a intermediate CoT state affect the correctness of the final output.
>
> ---
> **Q4: The justification for choosing the 85% reliability threshold in Figure 2b.**
>
> **R4:** We set this value as a threshold based on two considerations
> - In model-editing literatures, maintaining >85% utility is an empirical lower bound for a "non-destructive" edit.
> - In Fig 2b, the α values at 85% align with the inflection point: the plateau before the model's performance drops sharply.
>
> ---
>
> **Q5: BiCoT assumes the provider has white box access to the model, which seems inconsistent with Sec 2.1 that white box assumptions are often unrealistic.**
>
> **R5:** **The inconsistency arises from two distinct phases involving different parties**. In the embedding phase, the provider has white-box access to their own model during development. Sec 2.1 discusses the verification phase, where the model owner must authenticate whether a suspicious third-party's deployed model is stolen; here, white-box access to the suspect's system is unrealistic.
>
> ---
>
> **Q6: In Sec 4.4, it is unclear whether the variable v corresponds to the final target and how v is used during verification.**
>
> **R6:** v is the final geometric optimization target. The loss (Eq. 3) forces the hidden states of anchor tokens to collapse onto this fixed secret vector v during training. Since hidden states are inaccessible in black-box settings, v is mapped to the logit space via the language head $W_{head}v$. The verification score is the projection of the observed API logits onto this mapped signature (Eq. 6).
>
> ---
>
> **Q7: The specific layer from which the hidden state h is extracted and the justification for it.**
>
> **R7:** The hidden state $h$ is extracted from the 4th-to-last layer (L-4) (Appendix C.2/C.3; Tab 6, Line 850). The L-4 layer is chosen as it contains high-level, crystallized semantic information (Sec 3.3). This aligns with previous papers (Internvl, Chen, Zhe, et al., 2024).
>
> ---
>
> **Q8: The connection between attacks and the three categories in Sec 4.1 is unclear and the details are insufficient.**
>
> **R8:** Our evaluations (Sec 5.6) map to the threat model in Sec 4.1:
> * Input-level: We sample raw queries from GSM8K without modification, without any worry for input filtering.
> * Model-level: SFT, PEFT, and QT evaluate robustness against direct model weight modifications.
> * Output-level: This includes Gaussian noise and the adaptive ANCHOR attack.
>
> ---
>
> **Q9: The advantage of a higher MD in Tab 2 is unclear, given SEAL performs well despite a lower MD.**
>
> **R9:** Tab 2 evaluates a clean scenario. While AUC=1.0 shows separability, MD measures the "safety buffer" against attacks. A small MD (like SEAL's ~1.9) is easily bridged by adversarial shifts, causing detection failures. BiCoT's MD(11.67) guarantees robust verification under severe attacks, which is its core practical advantage.
>
> ---
>
> **Q10: Clarification of threat model, system roles, and the exact stage at which verification occurs.**
>
> **R10:** The workflow involves two main parties (Sec 4.1): The Provider embeds the watermark pre-release. The Attacker illegally downloads, fine-tunes, and deploys the model via a commercial API. Verification occurs post-deployment: if the Provider suspects theft, they query the Attacker's black-box API and analyze the returned logits to prove ownership.

---

> > ### Author Rebuttal · Reviewer_hdd8 · 2026-04-04
> >
> > Thank you for the rebuttal. However, my concerns are still not fully addressed.
> >
> > **Q2**. The paper still lacks a clear formal definition of critical tokens in the main text. It also remains unclear how this behavior is measured experimentally and whether it consistently holds beyond the reported case.
> >
> > **Q4**. The choice of the 85% threshold still appears largely empirical. I do not yet see a rigorous analysis showing why this value is appropriate or whether the conclusion is stable under other thresholds.
> >
> > **Q7**. The response states that the hidden state is taken from L-4, but it still does not explain why this layer is chosen. It is unclear whether this is the best layer in the current setting or how the layer should be selected in other settings.
> >
> > **Q8**. I am still confused about the mapping between the attacks in Section 5.6 and the threat categories in Section 4.1. In particular, the input level threat in the paper refers to prompt filtering of suspicious queries, but the response instead discusses using raw GSM8K queries without modification, which does not address that concern. I also still do not have enough implementation detail for the evaluated attacks

---

> > > ### Author Response · Authors · 2026-04-06
> > >
> > > Thanks for your time and giving us opportunities to clear your concerns!
> > >
> > > **Q2**. Lack a clear formal definition of critical tokens and is unclear how this behavior is measured and whether it consistently holds.
> > >
> > > R2:   Let the prompt be X, the generated trace be $S=(s_1, \ldots, s_N)$, and the canonical gold answer be Y. We score each token by leave-one-out removal under a fixed answer probe Q:$$Δ L = \log P(Y \mid X \oplus S \oplus Q) - \log P(Y \mid X \oplus S_{{\setminus i}} \oplus Q).$$
> > > A token is defined as critical if ΔL > 1. The added experiment on 12 model-dataset combinations each on 1000 samples results showed in tab below.
> > > |Model|Dataset|CoT Max ΔL|Answer Max ΔL|CoT Critical%|Answer Critical%|
> > > |-|-|-|-|-|-|
> > > |Qwen2.5-7B|gsm8k|0.357|1.784|0.5|23.1|
> > > ||svamp|0.060|2.586|0.1|28.3|
> > > ||commonsenseqa|0.047|1.055|0.1|13.8|
> > > ||arc_challenge|0.074|3.522|0.3|18.3|
> > > |Mistral-7B|gsm8k|0.477|2.724|0.3 |34.2|
> > > ||svamp|0.200|3.080|0.4|31.4|
> > > ||commonsenseqa|0.212|0.951|0.5|19.6|
> > > ||arc_challenge|0.322|2.619|0.6|22.3|
> > > |gemma-2-9b|gsm8k|0.104|2.271|0.0|29.6|
> > > ||svamp|0.037|2.435|0.0|29.5|
> > > ||commonsenseqa|0.010|1.475|0.0|30.1|
> > > ||arc_challenge|0.045|3.716|0.0|44.0|
> > > |All models|Overall|0.134|2.312|0.2|26.5|
> > >
> > > the answer span dominates the CoT span.  Answer Max ΔL = 2.31 versus CoT Max ΔL= 0.13, and the critical-token rate is 26.5% in the answer span versus only 0.2% in CoT. These results show that the evidence most critical for recovering the correct answer is concentrated in the final answer payload rather than broadly distributed throughout the generated CoT.
> > >
> > > **Q4**. Justification of the 85% threshold and an analysis of the robustness of the conclusions under alternative threshold
> > >
> > > R4: The role in our paper revisiting part is not to claim that 0.85 is uniquely correct, but to operationalize the existence of a stable regime where anchor perturbations remain non-disruptive. We add a new experiment on  tau in {0.75, 0.80, 0.85, 0.90, 0.95} on GSM8K/Alpaca(1000 test examples, 5 noise seeds) and evaluate conditional retained correctness, $P(EM(y_{\alpha}, y_{GT})=1| EM(y_0, y_{GT})=1)$. Tab below shows that retained correctness stays at 0.975-0.986 on average and 0.959-1.000 in the worst seed across all. Accordingly, the admissible interval remains [0, 6] for every tau up to 0.95. This confirms that our conclusion depends on the existence of a broad stable plateau, not on the specific numeric choice tau = 0.85; notably, the operating point used in our method (epsilon = 6.0) lies within this regime. Since even the worst seed remains above 0.95 for every tested alpha, the admissible interval is unchanged ([0, 6]) for all tau in [0.75, 0.95].
> > >
> > > |alpha|Conditional stability|Worst seed|Margin to tau=0.95|
> > > |-|-|-|-|
> > > |0|1.0 ± 0.0|1.0|+0.0500|
> > > |0.5|0.9753 ± 0.0103|0.9589|+0.0089|
> > > |1|0.9753 ± 0.0055|0.9726|+0.0226|
> > > |2|0.9781 ± 0.0110|0.9589|+0.0089|
> > > |3|0.9781 ± 0.0067|0.9726|+0.0226|
> > > |4|0.9863 ± 0.0087|0.9726|+0.0226|
> > > |6*|0.9808 ± 0.0164|0.9589|+0.0089|
> > >
> > > **Q7:** The response states that the hidden state is taken from L-4, but it still does not explain why this layer is chosen. It is unclear whether this is the best layer in the current setting or how the layer should be selected in other settings.
> > >
> > > R7:  Our original choice of L-4 was motivated by prior practice in papers (Internvl, Chen, Zhe, et al., 2024). We conducted a detailed layer ablation over candidate upper layers {-2, -4, -6} using the same procedure in the paper. We find that all three tested layers achieve saturated verification performance (AUC = 1.0 and pAUC@5%FPR = 1.0), indicating that the method is not sensitive to the exact choice within this upper-layer band. Therefore, the role of L-4 in our paper is not that of a brittle single-layer optimum.
> > >
> > > **Q8:** Clarification of the attack-threat mapping and additional implementation details for the evaluated attacks.
> > >
> > > R8: Sec. 4.1 defines a broad threat taxonomy, while Sec.5.6 evaluates a concrete set of attacks; they are related but not one-to-one. In particular, the input-level threat in Sec.4.1 refers to prompt filtering of suspicious verification queries, and this threat is not directly benchmarked in Sec. 5.6/Table 4. What we intended to convey is only that BiCoT uses ordinary task prompts rather than rare trigger prompts, so prompt filtering is less naturally applicable; this is a qualitative distinction, not an evaluated benchmark. The attacks in Sec. 5.6 map instead as follows: model-level attacks = Gaussian parameter noise, SFT, PEFT, MM, and QT; output-level attack = ANCHOR.
> > > The detailed attack information: QT uses 8-bit symmetric fake quantization; Gaussian noise adds parameter perturbations (std=0.05 in the main runner); MM is simulated as a merge with a drifted model via theta_wm + (1-alpha) * epsilon, with alpha=0.7; PEFT uses LoRA on q_proj/v_proj with r=8, lora_alpha=32, dropout=0.1; SFT is full-parameter fine-tuning for a 3 steps; and ANCHOR is a decoding-time token-suppression attack over digit anchors.

---

### Official Review · Reviewer_GmEy · 2026-03-18

**Soundness:** 2
**Presentation:** 1
**Significance:** 1
**Originality:** 2
**Overall Recommendation:** 4
**Confidence:** 2

**Summary:**

The paper seems to propose a method for model watermarking (i.e. protection of ownership of open-weight models by detecting their later unauthorized usage behind closed APIs). The method is injected via bi-level optimization that modifies the latent representations of the model during the CoT phase.

**Compliance With Llm Reviewing Policy:**

Affirmed.

**Final Justification:**

I have read the paper again and the other threads as well. I see that the paper is headed for acceptance and I have no good reason to block it, especially if the authors improve the writing and incorporate all changes from the rebuttals which seemed to have addressed many of the concerns. With all this in mind I've updated my score, but I remain with low confidence as I was unable to form an independent opinion here and had to rely on other reviewers and authors to get some understanding.

**Key Questions For Authors:**

/

**Limitations:**

I did not encounter either of these in my read.

**Strengths And Weaknesses:**

Despite my best effort and investing significantly more time into this paper than the others I have reviewed, I was unfortunately not able to understand the paper sufficiently to be able to confidently evaluate it. I find the paper to be written very poorly and mainly to consist of tortured overcomplicated sentences ("an effective watermark must inhabit this high-saliency subspace rather than the noise-tolerant margins") where key terms were never introduced; and grandiose claims that are not properly qualified ("BiCoT redefines the security boundaries of Large Language Models"). Importantly, the paper introduces many technical components quickly and through dense writing without giving proper motivation or background -- e.g., Fig 1-3 say a lot (saliency, capacity, diversity, importance, reasoning utility, signal propagation, linearity) but are very briefly explained with no details of the concrete methodology or meaning of what was measured. The issue of "drift" appears equally unmotivated and introduces propositions/theorems that are simply stated but not contextualized in a way that is understandable, making it hard to understand their soundness and value. I was equally unable to understand the setup of the experiments: it seems that the verifier observes the logits of the CoT process (which is highly unrealistic for this setting and would make the method not too significant), yet it claims to perform black-box detection. Regarding the theoretical contributions, 4.1 is a trivial identity, 4.2 seems tautological and 4.3 is simply stated with no proof or discussion. In conclusion, I have concerns about the rare parts of the paper that I managed to fully understand, and independently from that feel that the paper is not ready for publication in its current state of particularly bad presentation. To leave room for being wrong here I will mark my review with low confidence and make sure to read other reviewers' opinions in hopes that they were able to evaluate the paper more concretely.

---

> ### Author Rebuttal · Authors · 2026-03-31
>
> Thank you very much for the significant time and effort you invested in reviewing our paper. We sincerely apologize for the dense writing and the frustration it caused. We hope the following clarifications regarding the experimental setup, the concrete methodology of our foundational analysis, and the theoretical proofs can help alleviate your concerns.
>
> Q1: The assumption that the verifier observes the logits of the CoT process is highly unrealistic for a black-box setting, undermining the significance of the method.
>
> **R1:** We completely agree that the assumptions regarding API access deserve rigorous scrutiny, and we share your perspective on the importance of API practicality. As detailed in our response to Reviewer CL61 (Q1), our method does not require full-vocabulary logits. It relies strictly on standard Top-k logprobs, which is a universally accepted black-box protocol supported by major commercial APIs. We kindly invite you to review our response to Reviewer CL61 for our newly added API-limited experiments.
>
> Q2: The concrete methodology and meaning behind the metrics in Figures 1-3 (capacity, diversity, saliency, importance, linearity) are not properly explained.
>
> **R2:** We apologize for not thoroughly detailing these metrics in the main text. The formal mathematical definitions and experimental procedures are provided in Appendix C (Lines 625-696), which we will move to the main text to ensure clarity:
>
> - **Capacity:** Measured via the Effective Token Count, computed as the exponential of the Shannon entropy (Eq. 16). It quantifies the information density of the generated sequence.
> - **Diversity:** Measured via the Jaccard Similarity (Eq. 17) across independent generations from the same prompt. Lower similarity in CoT indicates greater intrinsic representational space.
> - **Importance:** Measured by randomly masking a ratio r of tokens and calculating the drop in the log-probability of the correct final answer (Eq. 18).
> - **Gradient Saliency:** Computed as the L2 norm of the gradient of the final-answer cross-entropy loss with respect to the hidden states of CoT tokens (Eq. 19).
> - **Linearity:** Quantified by the Pearson correlation coefficient between the injected signal strength and the L2 shift in output logits (Eq. 22).
>
> Q3: The issue of "drift" appears unmotivated, and the theoretical contributions (4.1, 4.2, 4.3) are simply stated without proofs, context, or discussion of their value.
>
> **R3:** We apologize for the lack of forward references to the proofs in the main text.
>
> - **Motivation for Drift:** "Drift" is highly motivated by real-world adversarial attacks. When an attacker steals a model, operations like Supervised Fine-Tuning (SFT) or Quantization (QT) cause the model's entire latent manifold to systematically shift (drift) in the vector space (Sec 5.6; Appendix E.2). If we do not account for this drift, simple detection thresholds will fail, leading to false accusations.
> - **Value of 4.1 and 4.2:** While Proposition 4.1 is a known geometric equivalence, stating it explicitly is necessary to guarantee that optimizing the L2 loss directly expands the cosine signal-to-noise margin. Proposition 4.2 formally ensures that our calibration mechanism successfully cancels out the systematic bias caused by the attacker's drift.
> - **Proof for Theorem 4.3:** Theorem 4.3 is not stated without proof; the rigorous mathematical derivation is fully detailed in Appendix A (Lines 550-604). This theorem is the cornerstone of our verifier, proving that the False Positive Rate (FPR) remains strictly bounded by the significance level α regardless of how heavily the attacker distorts (drifts) the model.
>
> Q4: Poor writing, undefined terms (e.g., "high-saliency subspace", "noise-tolerant margins"), and grandiose claims that are not properly qualified.
>
> **R4:** We deeply appreciate your candid feedback on the presentation. We acknowledge that our phrasing was overly dense and some claims were overstated. In the revised manuscript, we will:
>
> 1. **Define terms explicitly upon first use:** For example, we will clearly define the "high-saliency subspace" simply as the specific hidden dimensions of structural anchor tokens (e.g., digits) that exert the highest causal influence on the final answer, as calculated by their gradient norms.
> 2. **Tone down the language:** We will revise grandiose statements, such as replacing "BiCoT redefines the security boundaries..." in the conclusion with more grounded, objective language that strictly reflects our empirical findings.
> 3. **Restructure for readability:** We will significantly expand Sec 3 to properly contextualize Figures 1-3 within the main text, ensuring the narrative flows logically without requiring the reader to hunt for definitions.
>
> We hope these clarifications demonstrate the mathematical soundness and practical significance of our work, and we are fully committed to improving the paper's readability in the revision.

---

> > ### Author Rebuttal · Reviewer_GmEy · 2026-04-04
> >
> > Thank you for the rebuttal. The explanations are helpful but I would need to revisit the full paper and the other reviews to be able to form an educated opinion, for which I unfortunately did not have time before the rebuttal acknowlegment deadline. I assure the authors that I intend to do that before submitting my final review in the coming days.

---

> > > ### Author Response · Authors · 2026-04-07
> > >
> > > Dear Reviewer GmEy,
> > >
> > > Thank you so much for your candid update and for your continued dedication to reviewing our work. We deeply appreciate that you are willing to invest your valuable time in the coming days to revisit our full paper and the other reviewers' threads.
> > >
> > > Since the discussions across different threads have grown quite extensive, we wanted to provide a **quick navigation guide** to the most critical new experiments and data we provided to the other reviewers. We hope this digest saves your time and helps you easily locate the exact threads that address your initial concerns:
> > >
> > > **1. Practicality of the API Threat Model & Black-Box Logits**
> > > You rightly pointed out that full-logit access might be unrealistic. We fully agree and have addressed this extensively with Reviewer CL61:
> > >
> > > *   **Where to find it:** Please see our **"Rebuttal to Reviewer CL61" (Response R1)** for `new API-limited experiments` proving that observing only Top-k logprobs achieves AUC=1.0.
> > > *   **Where to find it:** Please see our follow-up **"Reply Rebuttal Comment" to Reviewer CL61** for a new `comprehensive survey table of current commercial/open-source APIs(OpenAI, Gemini, Qwen, etc.) and open-source engines(vLLM, TGI)`, demonstrating that exposing Top-$k$ logprobs is the industry standard, not a niche feature.
> > >
> > > **2. Geometric Separation and Text Generation Quality**
> > > To clarify how we maintain generation quality without interfering with semantic text, we provided a microscopic, token-level geometric analysis showing the rapid exponential decay of the watermark signal.
> > >
> > > *   **Where to find it:** Please see our **"Rebuttal to Reviewer eD21" (Response R1)** for the `detailed token-level trajectory tables ("Causal Halo Effect")`.
> > > *   *Note: We are very encouraged that this specific token-level evidence fully resolved Reviewer eD21's concerns, leading them to raise their final score to 5 (Accept).*
> > >
> > > **3. Generalization Beyond Math (Text, Code, Law)**
> > > To prove our framework isn't heavily tailored to digits, we conducted extensive Cross-Domain Generalization Experiments (CommonsenseQA, LexGLUE, HumanEval).
> > >
> > > *    **Where to find it:** Please see our **"Rebuttal to Reviewer CL61" (Response R2)** for `the complete table showing AUC=1.0` and negligible accuracy drops across these non-mathematical domains.
> > >
> > > **4. Robustness of Revisiting Section and Formal Definitions**
> > > We provided additional formal definitions and large-scale ablations regarding our threshold choices and "critical tokens."
> > >
> > > *   **Where to find it:** Please see our latest **"Reply Rebuttal Comment" to Reviewer hdd8 (Responses R2 & R4)** for the `new 12 model-dataset combination evaluation` (on 12,000 samples) and the threshold stability tables.
> > >
> > > We sincerely hope this structured roadmap serves as a helpful tool as you navigate the OpenReview threads to form your final educated opinion. Please do not hesitate to let us know if you have any further questions!
> > >
> > > Thank you once again for your hard work and fair judgment.
> > >
> > > Best regards,
> > >
> > > Authors of Submission 16946

---

### Decision · Program_Chairs · 2026-04-30

**Decision:**

Accept (regular)

**Comment:**

2x weak accept, 1x weak reject, 1x accept. This paper proposes a watermarking framework for large language models that embeds ownership signatures into chain-of-thought reasoning representations and verifies them through a drift-robust detector designed for black-box model-stealing settings. The reviewers agree on the (1) originality of using chain-of-thought as the watermark carrier, (2) strong empirical detection performance and robustness across multiple attacks, and (3) the potential value of the drift-correction verifier and accompanying ablations. However, they note (1) practical limitations stemming from the reliance on logit access and the boundary case where logits or CoT are unavailable, (2) remaining questions about generalization beyond math-style reasoning and digit anchors despite added rebuttal evidence, and (3) presentation and rigor issues in definitions, threat-model clarity, and some methodological explanations, with one reviewer remaining unconvinced even after rebuttal. The authors’ follow-up responses and added experiments appear to have addressed most concerns for two reviewers and substantially clarified the method, although some practical and exposition issues remain, so the AC leans to accept this submission.